# Kinesin motility is driven by subdomain dynamics

Wonmuk Hwang[1,2,3]*, Matthew J Lang[4,5]*, Martin Karplus[6,7]*

[1]Department of Biomedical Engineering, Texas A&M University, College Station, United States; [2]Department of Materials Science & Engineering, Texas A&M University, College Station, United States; [3]School of Computational Sciences, Korea Institute for Advanced Study, Seoul, Korea; [4]Department of Chemical and Biomolecular Engineering, Vanderbilt University, Nashville, United States; [5]Department of Molecular Physiology and Biophysics, Vanderbilt University School of Medicine, Nashville, United States; [6]Department of Chemistry and Chemical Biology, Harvard University, Cambridge, United States; [7]Laboratoire de Chimie Biophysique, ISIS, Université de Strasbourg, Strasbourg, France

**Abstract** The microtubule (MT)-associated motor protein kinesin utilizes its conserved ATPase head to achieve diverse motility characteristics. Despite considerable knowledge about how its ATPase activity and MT binding are coupled to the motility cycle, the atomic mechanism of the core events remain to be found. To obtain insights into the mechanism, we performed 38.5 microseconds of all-atom molecular dynamics simulations of kinesin-MT complexes in different nucleotide states. Local subdomain dynamics were found to be essential for nucleotide processing. Catalytic water molecules are dynamically organized by the switch domains of the nucleotide binding pocket while ATP is torsionally strained. Hydrolysis products are 'pulled' by switch-I, and a new ATP is 'captured' by a concerted motion of the α0/L5/switch-I trio. The dynamic and wet kinesin-MT interface is tuned for rapid interactions while maintaining specificity. The proposed mechanism provides the flexibility necessary for walking in the crowded cellular environment.
DOI: https://doi.org/10.7554/eLife.28948.001

*For correspondence:
hwm@tamu.edu (WH);
matt.lang@vanderbilt.edu (MJL);
marci@tammy.harvard.edu (MK)

Competing interests: The authors declare that no competing interests exist.

## Introduction

Kinesin is an ATPase motor protein that walks along microtubules (MTs), to carry out vital functions, which include intracellular transport and cell division (*Vale, 2003*; *Hirokawa and Noda, 2008*). As the smallest known motor that can walk processively, it also serves as the canonical motor protein (*Block, 2007*; *Hwang and Lang, 2009*). Kinesin families use variations in subdomains to harness nucleotide-dependent conformational changes of the conserved motor head to generate diverse motility characteristics (*Cochran, 2015*), such as: direction reversal (*Endow and Waligora, 1998*; *Endres et al., 2006*; *Yamagishi et al., 2016*), MT polymerization/depolymerization (*Wordeman, 2005*; *Hibbel et al., 2015*), and motility with only a single head (*Kikkawa et al., 2000*).

To understand the mechanisms of the different kinesins, it is important to search for and elucidate the conserved features of the motor head that are involved in the nucleotide processing events of the motility cycle; that is, ATP binding, hydrolysis, and product release (ADP and $P_i$, inorganic phosphate) (*Figure 1A*). One of the most intensely studied family member is Kinesin-1 (Kin-1; hereafter we refer Kin-1 as kinesin). It forms a dimer to walk toward the MT plus-end using one ATP per step (*Figure 1B*) (*Svoboda et al., 1993*; *Block, 2007*). It unbinds from the MT in the ADP state, and after making a step, it releases ADP and enters the nucleotide-free APO state with high MT-affinity (*Cross, 2016*; *Hancock, 2016*). Binding of an ATP triggers forward force generation (the 'power

**eLife digest** Motor proteins called kinesins perform a number of different roles inside cells, including transporting cargo and organizing filaments called microtubules to generate the force needed for a cell to divide. Kinesins move along the microtubules, with different kinesins moving in different ways: some 'walk', some jump, and some destroy the microtubule as they travel along it. All kinesins power their movements using the same molecule as fuel – adenosine triphosphate, known as ATP for short.

Energy stored in ATP is released by a chemical reaction known as hydrolysis, which uses water to break off specific parts of the ATP molecule. The site to which ATP binds in a kinesin has a similar structure to the ATP binding site of many other proteins that use ATP. However, little was known about the way in which kinesin uses ATP as a fuel, including how ATP binds to kinesin and is hydrolyzed, and how the products of hydrolysis are released. These events are used to power the motor protein.

Hwang et al. have used powerful computer simulation methods to examine in detail how ATP interacts with kinesin whilst moving across a microtubule. The simulations suggest that regions (or 'domains') of kinesin near the ATP binding site move around to help in processing ATP. These kinesin domains trap a nearby ATP molecule from the environment and help to deliver water molecules to ATP for hydrolysis. Hwang et al. also found that the domain motion subsequently helps in the release of the hydrolysis products by kinesin.

The domains around the ATP pocket vary among the kinesins and these differences may enable kinesins to fine-tune how they use ATP to move. Further investigations will help us understand why different kinesin families behave differently. They will also contribute to exploring how kinesin inhibitors might be used as anti-cancer drugs.

DOI: https://doi.org/10.7554/eLife.28948.002

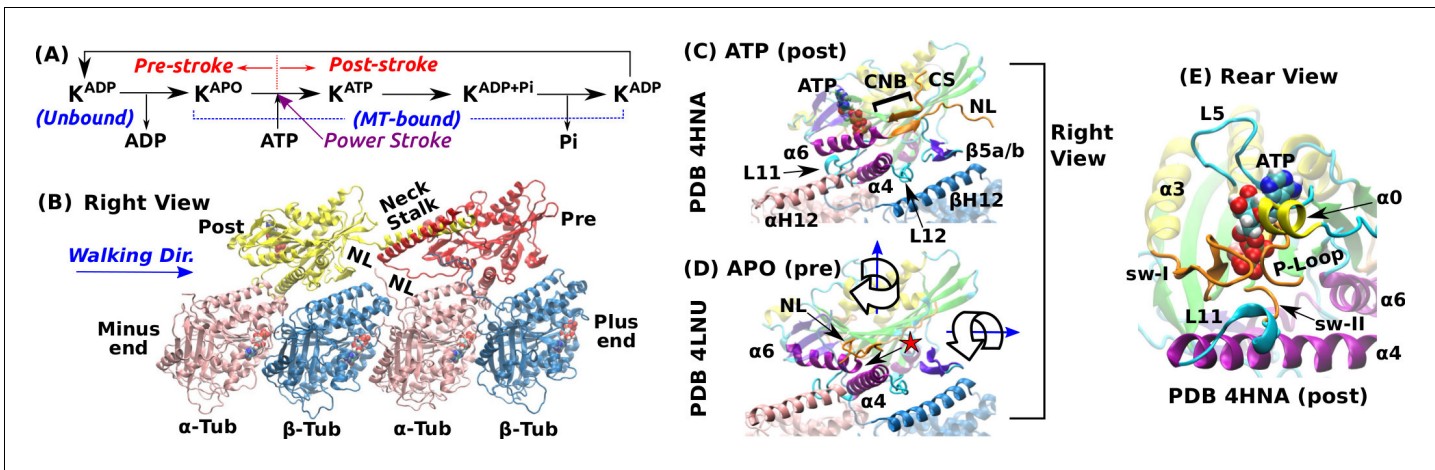

**Figure 1.** Overview of kinesin structure and motility cycle. (A) Diagram of the ATPase cycle of a motor head. Binding of an ATP triggers the conformational change to the post-stroke state. (B) Model of a kinesin dimer bound to the MT. The rear and front heads are in the post- and pre-stroke states, respectively. The neck linker (NL) connects the C-terminal end of the motor head to the $\alpha$-helical neck stalk. (C,D) Comparison between (C) post- (*Gigant et al., 2013*) and (D) pre-stroke (*Cao et al., 2014*) states, defined based on the orientation of $\alpha6$ relative to $\alpha4$. CS: cover strand. $\alpha H12/\beta H12$: C-terminal helices of $\alpha$-/$\beta$-tubulins that form major contacts with kinesin, mainly with L11, $\alpha4$, L12, and $\beta5a/b$ (also called L8). In the pre-stroke state, $\alpha6$ shortens and its C-terminal end connecting to the NL is positioned behind $\alpha4$ (red star). This is coupled with the rightward tilting and clockwise rotation about the vertical axis of the motor head (wide arrows). (E) The ATP binding pocket. MT is not shown. Kinesin structures are compared in *Supplementary file 1*. A complete list of kinesin domain names are in *Figure 1—figure supplement 1*.

DOI: https://doi.org/10.7554/eLife.28948.003

The following figure supplement is available for figure 1:

**Figure supplement 1.** Structural overview of kinesin.

DOI: https://doi.org/10.7554/eLife.28948.004

stroke'; *Figure 1A*) by driving the cover strand (CS) and the neck linker (NL), which are located respectively on the N- and C-terminal ends of the motor head, to fold into a $\beta$-sheet named the cover-neck bundle (CNB; *Figure 1C*) (*Rice et al., 1999*; *Hwang et al., 2008*; *Khalil et al., 2008*). ATP hydrolysis completes a step (*Milic et al., 2014*; *Andreasson et al., 2015*).

The *pre-* and *post-stroke* states differ in the motor head orientation relative to the MT. In the pre-stroke state, the head tilts rightward relative to the MT plus-end direction, and rotates clockwise when viewed from above (wide arrows in *Figure 1D*). The head rotates in the opposite direction in the post-stroke state (*Sindelar and Downing, 2010*).

The nucleotide pocket on the rear-left side of the motor consists of the phosphate loop (P-loop), switch-I (sw-I), and switch-II (sw-II) (*Figure 1E*). These elements are conserved among different proteins including myosin and G-protein (*Vale and Milligan, 2000*). Compared to an isolated kinesin, a MT-bound kinesin has an at least 10-fold higher ATP hydrolysis rate (*Vale, 1996*; *Ma and Taylor, 1997*), which suggests that the nucleotide pocket is allosterically controlled by the interface with the MT. Among kinesin's MT-facing domains (*Figure 1C*), L11 and $\alpha 4$ undergo large conformational changes upon binding to the MT. L11 is located after sw-II (*Figure 1E*), followed by $\alpha 4$ that N-terminally extends by a few turns when the motor head binds to the MT (*Supplementary file 1*) (*Sindelar and Downing, 2010*; *Atherton et al., 2014*; *Shang et al., 2014*). However, substantial conformational variations are present in these conserved domains, notably in sw-I and L11 (see *Supplementary file 1*). Also, the extent of the kinesin-MT interface varies depending on experimental conditions (*Morikawa et al., 2015*).

Thus, it is necessary to identify core features of the motor head that are essential for nucleotide processing. Such information about a single head is a prerequisite for the atomic-level understanding of the motility of a dimer. We characterize these features via multi-microsecond molecular dynamics simulations on the Anton supercomputer (*Shaw et al., 2009*; *Shaw et al., 2014*) of a motor head complexed with a tubulin dimer. Compared to previous all-atom simulations that used biasing potentials and were limited in time (*Li and Zheng, 2012*; *Shang et al., 2014*; *Chakraborty and Zheng, 2015*), the unbiased simulations described here reveal the conformational changes of kinesin-MT complexes on a more realistic time scale. We find that the nucleotide binding pocket is conformationally the most dynamic part of the motor head, whose internal motions actively drive the nucleotide processing events. In particular, we show how ATP hydrolysis occurs in a fluctuating environment, and demonstrate the role of the kinesin-MT interface for this process.

The dynamic nature of the kinesin mechanism elucidates how it robustly carries out its motility cycle despite significant conformational perturbations to the motor head and the MT in the crowded cellular environment (*Leduc et al., 2012*). The source of the chemical energy and the mechanism involved in 'walking on tracks' are applicable to other translocating motors such as myosin on actin (*Vale and Milligan, 2000*; *Hwang and Lang, 2009*), making the present mechanistic results of general interest.

## Results

### Simulation overview

We studied Kin-1 in different nucleotide or structural states. The names of the simulated systems are given below in italics. Simulation times and conformational states are in parentheses.

1. *ATP* (4.16 $\mu$s, post-stroke): ATP-kinesin bound to the MT (*Figure 1C*).
2. *Kin-only* (5.35 $\mu$s; post-stroke): ATP-kinesin without MT.
3. *ADP+Pi* (5.56 $\mu$s; post-stroke): State after ATP hydrolysis, with ADP and P$_i$.
4. *ADP$_{pre}$* (2.91 $\mu$s; pre-stroke): State immediately before ADP release upon binding to the MT.
5. *APO$_\alpha$* (1.19 $\mu$s; pre-stroke): APO state. Sw-I forms an $\alpha$-helix (*cf.*, *Supplementary file 1*).
6. *APO* (4.00 $\mu$s; pre-stroke): APO state. Sw-I is disordered (*Figure 1D*).

We also carried out simulations of Eg5 (Kin-5 family). However, currently only Kin-1 has atomic-resolution x-ray structures of the motor head complexed with the MT in both the pre- and post-stroke states (*Gigant et al., 2013*; *Cao et al., 2014*). Consequently, we focus our analysis on Kin-1, and use Eg5 for comparison.

## Functionally important subdomains are mobile

We quantified the conformational motion by measuring the average displacement and root-mean-square deviation (RMSD) of $C_\alpha$ atoms relative to the first frame, during the first and the last 400 ns (*Figure 2A,B* and *Figure 2—figure supplement 1A–D*). Displacements represent deformation from the initial structure, and RMSD shows the degree of conformational fluctuation. To find conformational changes over time, we plotted rolling averages of mean $C_\alpha$ displacements that are greater than 1 Å (*Figure 2C*). To focus on changes in the core domains, the CS and NL, located at the termini of the motor head, were excluded from this calculation. Displacements saturate after 1–3 $\mu$s. *Kin-only* exhibits the greatest displacement, reflecting larger changes without the MT. $ADP_{pre}$ had a large displacement between 2–3 $\mu$s, which is due to the release of ADP described below. Average displacements of other MT-bound structures were in the 2.5–3 Å range.

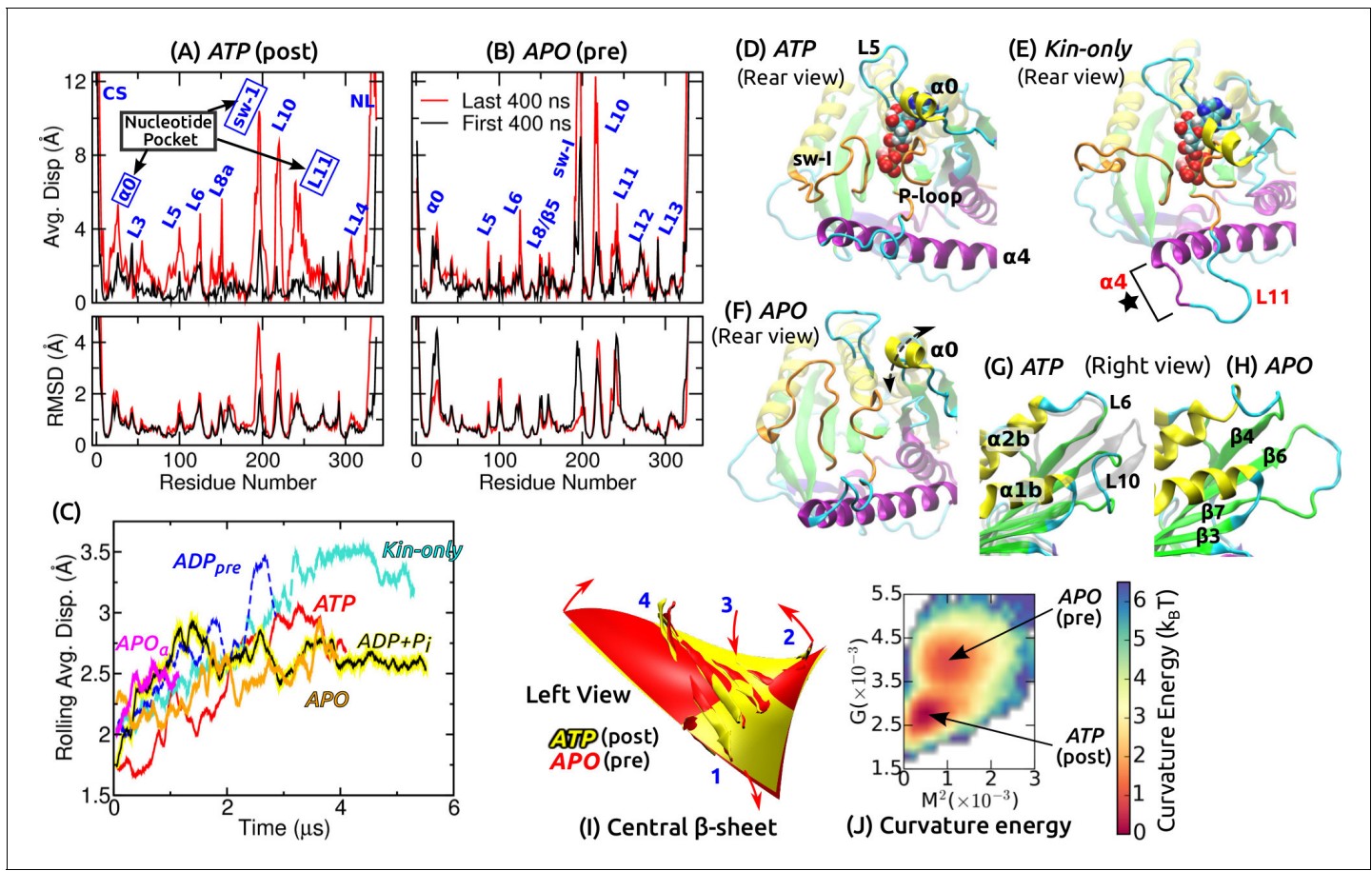

**Figure 2.** Conformational dynamics of the motor head. Related data for systems not shown here are in *Figure 2—figure supplement 1*. (A,B) Average displacement of $C_\alpha$ atoms from their initial positions (top row) and RMSD (bottom row) during the first/last 400 ns (black/red lines). The central $\beta$-sheet of kinesin was used for alignment. Domain names are in *Figure 1—figure supplement 1*. (A) *ATP* and (B) *APO*. (C) Rolling average (100-ns window) of displacements of $C_\alpha$ atoms excluding the CS and NL. Only residues with displacements greater than 1 Å were considered. (D–F) Conformation of domains around the nucleotide pocket at the end of simulation (cf., *Figure 1E*). (D) *ATP*. Sw-I lost its pseudo $\beta$-hairpin conformation. (E) *Kin-only*. N-terminal end of $\alpha$4 (star) and L11 also unfolded. (F) *APO*. Direction of motion of $\alpha$0 is marked by a dashed arrow. (G,H) Unfolding of the front end of the motor head. (G) *ATP*. The initial structure is shown in transparent gray, for comparison. (H) *APO*. (I) Curvature of the central $\beta$-sheet in *ATP* (yellow) and *APO* (red). Surfaces take the average curvatures in respective simulations. A higher saddle-point (Gaussian) curvature in *APO* can be seen by the greater bending of the corners 1, 3 and 2, 4 in opposite directions (arrows). (J) Combined energy landscape parameterized by the mean ($M^2$) and Gaussian ($G$) curvatures of the central $\beta$-sheet.

DOI: https://doi.org/10.7554/eLife.28948.005

The following figure supplement is available for figure 2:

**Figure supplement 1.** Conformational behavior of the motor head.
DOI: https://doi.org/10.7554/eLife.28948.006

Large displacements other than the CS and NL are localized around the nucleotide pocket (sw-I, L11, $\alpha0$) and the front end of the motor head (L10) (*Figure 2D–H* and *Figure 2—figure supplement 1E–J*). Sw-I (R190–S204; *Figure 1—figure supplement 1*) is particularly flexible. Only in *APO$_\alpha$*, sw-I maintained the initial $\alpha$-helical conformation with low displacement (*Figure 2—figure supplement 1D,F*), which could be due to its relatively short simulation time. Sw-I in its hairpin-like state was also mobile during the first 400 ns (*Figure 2A*), which is consistent with a previous 400-ns simulation study (*Chakraborty and Zheng, 2015*). L11 initially adopts an $\alpha$-helical turn in all systems except for *APO* (*Figure 1E*). It unfolded in *APO$_\alpha$*, becoming similar to *APO* (*Figure 2—figure supplement 1F*). In addition to the MT, a nucleotide may thus be needed to stabilize the $\alpha$-helical turn in L11 (*Yamada et al., 2007*). In *Kin-only*, L11 and the N-terminal part of $\alpha4$ unfolded (*Figure 2E*, star), which agrees with available x-ray structures of kinesin without MT (*Supplementary file 1*).

The adenosine group of the nucleotide is close to $\alpha0$ and L5 (*Figure 1E*). In Kin-5, L5 is about 17–21-aa long and exhibits large nucleotide-dependent conformational changes (*Behnke-Parks et al., 2011*; *Goulet et al., 2012*; *Goulet et al., 2014*). In Kin-1, it is 9-aa long and fluctuates less compared to $\alpha0$ (*Figure 2A,B* and *Figure 2—figure supplement 1A–D*). $\alpha0$, which has not been considered previously, fluctuates mostly up-and-down (arrow in *Figure 2F*). Below, we show that its mobility aids in binding of ATP.

The front end of the motor head (especially L10) also exhibits large deformation and fluctuation (*Figure 2G,H* and *Figure 2—figure supplement 1G–J*). This region has a high temperature factor, is deformed, or exhibits low electron density in several Kin-1–MT structures (*Cao et al., 2014*; *Atherton et al., 2014*; *Shang et al., 2014*) and also in Kin-14 (*Hirose et al., 2006*). The front end interacts with the C-terminal tail of a full-length kinesin when it is in an auto-inhibited state (*Kaan et al., 2011*). Its compliance may thus be more relevant to tail binding rather than nucleotide processing (*Verhey and Hammond, 2009*). Another domain possessing relatively high flexibility is L8/$\beta5$ on the frontal side of the interface with the MT (*Figure 1C*), whose interaction with the MT varies (*Atherton et al., 2014*; *Shang et al., 2014*; *Morikawa et al., 2015*). In *Kin-only*, L12 facing the MT also shows a large displacement, as expected without the MT (*Figure 2—figure supplement 1A*).

## Kinesin's central $\beta$-sheet does not store enough energy to drive nucleotide processing

The curvature of the central $\beta$-sheet is another aspect of kinesin's conformation. For the evolutionarily related myosin (*Vale and Milligan, 2000*), the corresponding $\beta$-sheet in the transducer domain exhibits large, nucleotide-dependent curvature changes (*Coureux et al., 2004*). Its deformational energy has been speculated to partly drive force generation in myosin (*Sweeney and Houdusse, 2010*). For kinesin, the role of $\beta$-sheet curvature has been debated (*Arora et al., 2014*; *Atherton et al., 2014*; *Shang et al., 2014*).

For each coordinate frame, we measured the mean curvature $M^2$ (concaveness) and the Gaussian curvature $G$ (saddle-point curvature) of the central $\beta$-sheet, and calculated the curvature free energy (potential of mean force; PMF) for each simulation (described in Materials and methods). These two curvatures quantify the bending and twisting of the $\beta$-sheet, respectively (*Sun et al., 2003*). Pre-stroke states had generally higher curvature, especially in $G$ (*Figure 2I* and *Figure 2—figure supplement 1K*). Since *ATP* and *ADP+Pi* have very similar curvature, neither ATP hydrolysis nor $P_i$ release (see below) is driven by the tendency of the $\beta$-sheet to adopt a higher curvature. Similarly, *ATP* and *Kin-only* had nearly the same curvature, indicating that binding to the MT does not impose any strain on the central $\beta$-sheet (*Figure 2—figure supplement 1K*).

We obtain information concerning the effect of curvature changes between pre- and post-stroke states by superposing the PMFs for *ATP* and *APO* (*Figure 2J*). *APO* has a local free energy minimum that is 0.84 $k_BT$ ($k_BT$: thermal energy at 300 K) higher than that of *ATP*. There is also a ~1.7 $k_BT$ energy barrier from *ATP* towards *APO*. The pre- and post-stroke states respectively have similar PMFs regardless of the details of individual simulations (*Figure 2—figure supplement 1K*). Further, the PMF in *Figure 2J* does not directly represent the properties of the central $\beta$-sheet itself, but it implicitly reflects the energetics of the whole system, including domains surrounding the $\beta$-sheet, nucleotide and the MT, in controlling the curvature. These free energies are well below the 10-$k_BT$ free energy (8 nm step×5 pN stall force) used by kinesin, which can also be seen by the large overlap in individual curvature distributions between the pre- and post-stroke states (*Figure 2—figure*

*supplement 1K*). By comparison, the rotary motor $F_1$-ATPase has about 5-$k_BT$ curvature energy changes (*Sun et al., 2003*). Therefore, curvature changes in kinesin are not substantial enough to drive ATP hydrolysis nor the transitions between pre- and post-stroke states.

## Kinesin-MT interface is dynamic and hydrated

Next we studied the motion of the motor head relative to the MT. For positional and orientational reference, we used the central $\beta$-sheet, $\alpha6$, $\alpha4$, and $\beta5a/b$ (*Figure 3A* and *Figure 3—figure supplement 1A*). The central $\beta$-sheet, with its low RMSD, represents the overall position and orientation of the motor head. $\alpha6$ changes its orientation between pre- and post-stroke states (*Figure 1C,D*). $\alpha4$ and $\beta5a/b$ are MT-binding domains. For each domain, translations in longitudinal, transverse, and normal (perpendicular to the MT surface) directions, and rotations about these three directions were measured.

Translational and orientational changes between pre- and post-stroke states captured various aspects of available x-ray and cryo-EM structures of kinesin-MT complexes. The central $\beta$-sheet shifts mostly leftward in the post-stroke state (*Figure 3A*), and in all states, it fluctuates more in the

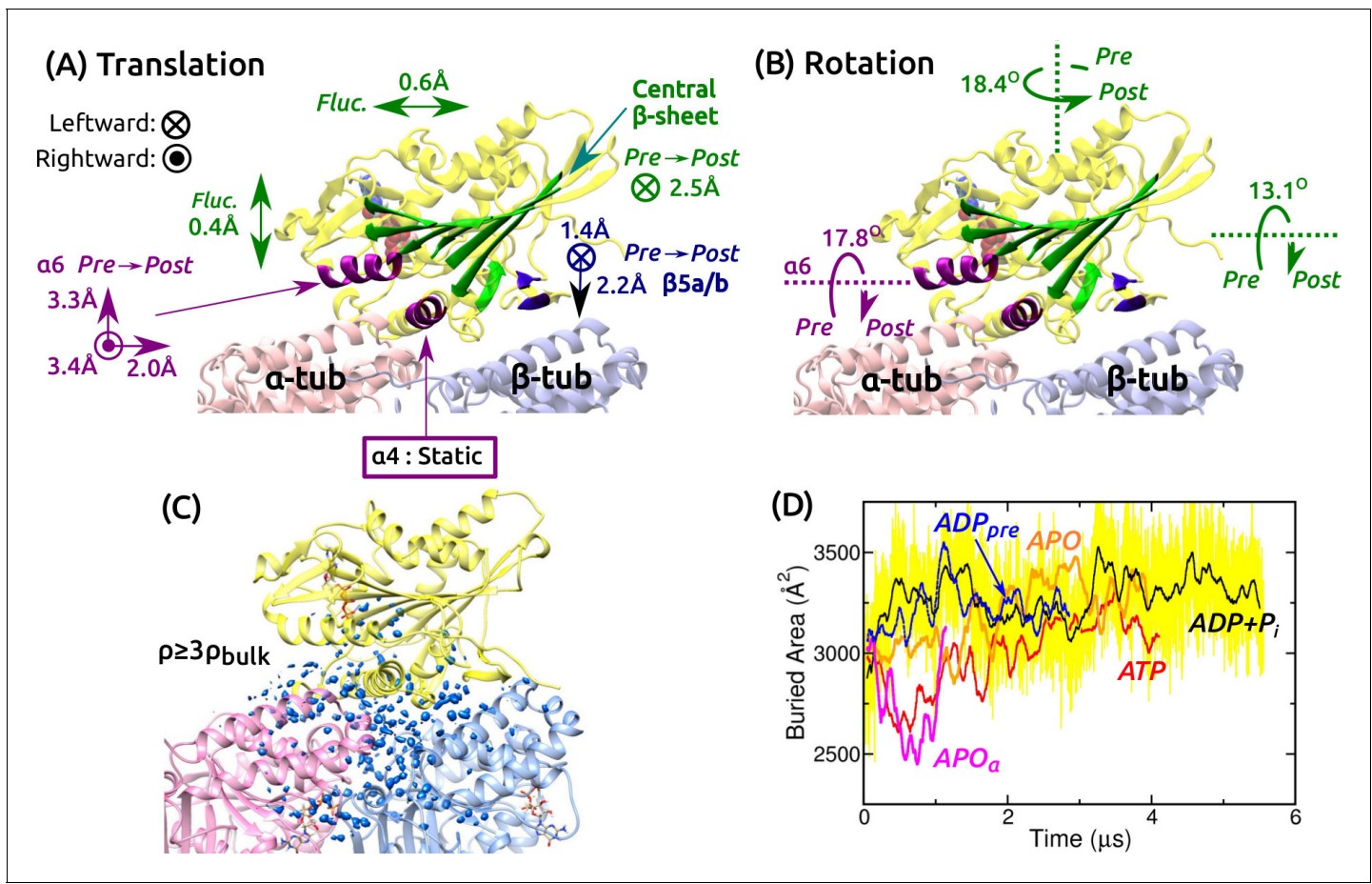

**Figure 3.** Mobility of kinesin on the MT. (A,B) Summary of translational and rotational motion. See *Figure 3—figure supplement 1A–G* for detailed analysis. Direction and magnitude of (A) translation and (B) rotation of major domains between pre- and post-stroke states. (C) Hydration of the kinesin-MT interface. Blue blobs: regions where water density is higher than three times the bulk value. The hydration shell at cutoff equal to the bulk density is in *Figure 3—figure supplement 1H*. (D) Trajectory of buried area between kinesin and the MT (100-ns rolling average). Yellow: raw data for *ADP+Pi*, revealing large fluctuation. Raw data for other simulations fluctuate with comparable magnitude.

DOI: https://doi.org/10.7554/eLife.28948.007

The following figure supplement is available for figure 3:

**Figure supplement 1.** Flexibility of the kinesin-MT interface.

DOI: https://doi.org/10.7554/eLife.28948.008

transverse direction, indicating an anisotropic compliance (*Figure 3—figure supplement 1B*). The shift in $\alpha6$ between pre- and post- stroke states agrees with its C-terminal end moving over $\alpha4$ (*Figure 1C,D* vs. *Figure 3A*; *Figure 3—figure supplement 1C*). $\alpha4$ is nearly stationary, so that it serves as an anchor for binding to the MT (*Figure 3—figure supplement 1D*). The vertical shifts of $\beta5a/b$ (*Figure 3A* and *Figure 3—figure supplement 1E*) have been observed in cryo-EM structures of kinesin-MT complexes in both APO and ATP-analog states, depending on experimental conditions (*Atherton et al., 2014*; *Morikawa et al., 2015*). The shifts are thus likely non-essential for the operation of kinesin. The central $\beta$-sheet and $\alpha6$ rotate as observed in crystal structures (*Figure 1C, D* vs. *Figure 3B* and *Figure 3—figure supplement 1F,G*).

We also calculated the water density map for the kinesin-MT interface during the last 500 ns. The map was visualized with two different density cutoffs. When a cutoff equal to the bulk density ($0.0333$ Å$^{-1}$) is used, globular hydration shells surround the interface (*Figure 3—figure supplement 1H*). With a cutoff equal to three times the bulk density, a collection of blobs appear, which correspond to regions where water oxygens are found with high probability during the simulation (*Figure 3C*). They are located within the kinesin-MT interface and crevices, for all simulations. Lack of any correlation between the extent of interfacial hydration and the conformational state can also be seen by the buried area within the kinesin-MT interface. It undulates with 80–720-ns correlation times and with instantaneous fluctuations of a few hundred Å$^2$ (e.g., yellow trace in *Figure 3D*).

We measured the binding energy between kinesin and the MT during the last 500 ns (*Figure 3—figure supplement 1I–L*). Pre-stroke states interact less with the $\beta$-tubulin (*Figure 3—figure supplement 1L*, squares), which is consistent with its front side ($\beta5a/b$) lifting from the MT (*Figure 3A*). Interaction with $\alpha$-tubulin differs more across simulations. Overall, *ATP* and *APO* have the strongest binding energy (*Figure 3—figure supplement 1L*, circles), which are in line with experiments where the ATP and APO states have high MT affinity compared to the ADP state (*Woehlke et al., 1997*). However, our binding energies do not include water-mediated interactions and entropic contributions, which are expected to be comparable to the binding energy in magnitude (*Zoete et al., 2005*), so that the net binding free energy is much smaller than those in *Figure 3—figure supplement 1L*. Thus, the calculated binding energies, although they reflect the interaction between kinesin and the MT in different nucleotide states, do not correspond quantitatively to the experimental binding affinities. In any case, the mobility of the motor head and extensive hydration of the interface observed in all simulations suggest that the kinesin-MT interface is highly dynamic. This point is further explored in the analysis of kinesin-MT contacts below.

## Nucleotide pocket experiences large changes in intra-kinesin contacts

To understand the conformational behavior of the system at the individual amino acid level, we traced all intra-kinesin and kinesin-MT contacts. Hydrogen bonds (H-bonds; including salt bridges) and nonpolar contacts were considered, majority of which form and break with less than 100% occupancy during the simulation (example occupancy trajectories are in *Figure 4—figure supplement 1A*). Among intra-kinesin contacts, there were fewer H-bonds (1100–1500) than nonpolar contacts (1800–2400). The occupancy distribution is U-shaped in logarithmic scale, majority of which have lower than 20% occupancy (*Figure 4A*). The number of contacts with greater than 80% occupancy were 137–153 (H-bond) and 302–339 (nonpolar). Among contacts showing irreversible transitions, we monitored those whose occupancy before breakage or after formation is greater than 80% (*Figure 4B*; *Supplementary file 2*). Post-stroke states had more contacts break than form, which occurred mainly within the first 3 $\mu$s (*Figure 4—figure supplement 1B*). Locations at which changes occurred are clustered around the nucleotide pocket and the front side (*Figure 4B*), that also had high RMSD (*Figure 2A*). With a bound nucleotide, contacts involving sw-I undergo extensive changes, which is responsible for the greater number of changes in the post- than in the pre-stroke state (*Figure 4—figure supplement 1B*). This is mainly because in the post-stroke state, contacts between the N- and C-terminal sides of sw-I forming the hairpin break. But contacts between sw-I with ATP and sw-II, necessary for the hydrolysis of ATP, remain intact (*Supplementary file 2A–C*; see below).

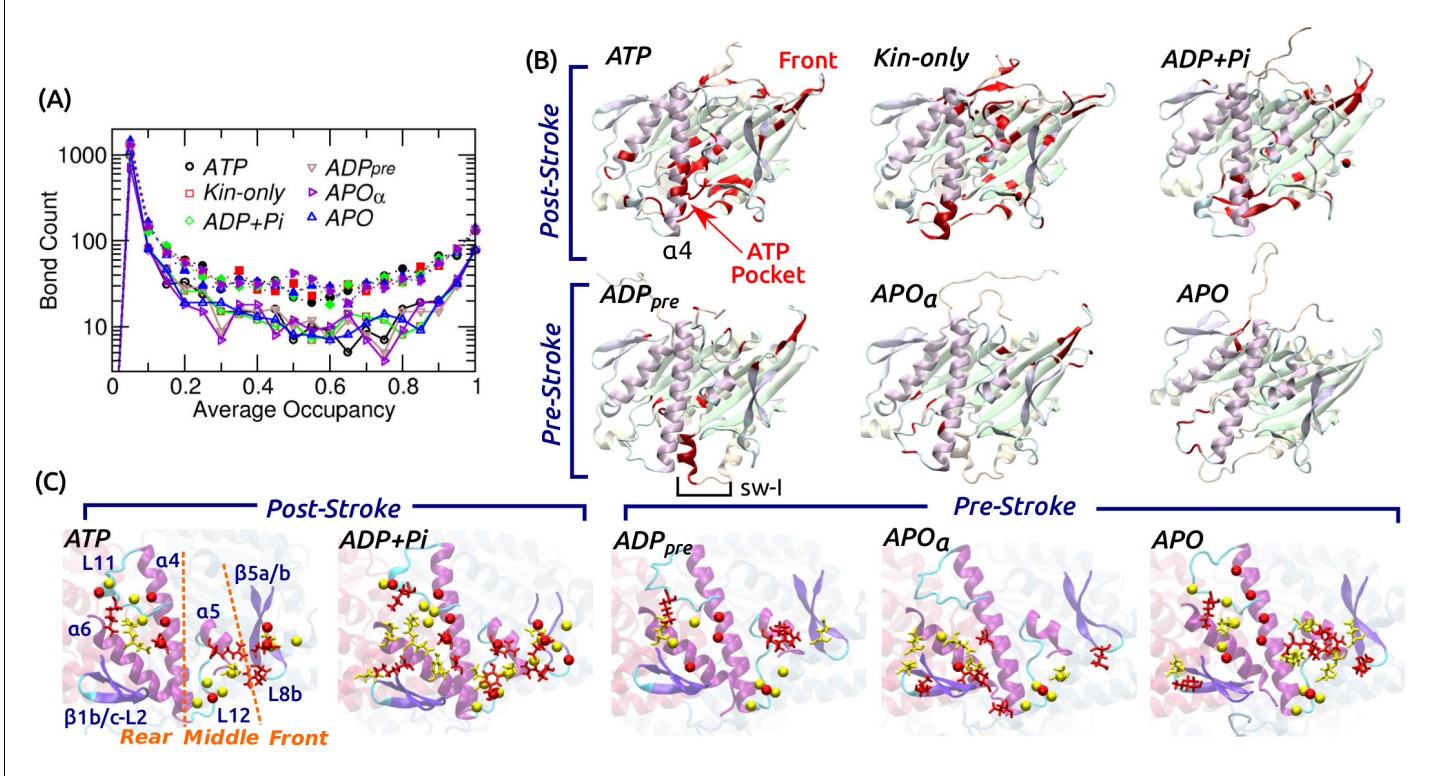

**Figure 4.** Contact analysis. (A) Occupancy distribution of intra-kinesin contacts. Filled symbols: nonpolar contacts. Open symbols with solid lines: H-bonds. *Figure 4—figure supplement 1A* shows examples of occupancy trajectories. (B) Locations of intra-kinesin contacts that broke or formed during simulation (colored red). Bottom view. See *Supplementary file 2* for the list of contacts and their transition times. Cumulative numbers of contacts formed or broken over time are in *Figure 4—figure supplement 1B*. Post-stroke states involve more contacts formed or broken, mainly around the ATP pocket and the front end of the motor head. The first frame of each simulation was used for visualization. In *APO*, the frontal part of the motor head was unstable from the beginning of the simulation, thus no clear contact changes were identified. (C) Residues forming kinesin-MT contacts with higher than 80% occupancy. Top view. For kinesin, only MT-facing domains are shown. Red: kinesin residues, yellow: MT residues. Stick: residues forming H-bonds, sphere: residues forming nonpolar contacts. In *ATP*, the three contact positions (rear, middle, front) are marked. Post-stroke states have more contacts with the front part. *Supplementary file 3* lists individual kinesin-MT contacts.

DOI: https://doi.org/10.7554/eLife.28948.009

The following figure supplement is available for figure 4:

**Figure supplement 1.** Dynamics of intra-kinesin and kinesin-MT contacts.

DOI: https://doi.org/10.7554/eLife.28948.010

## Kinesin-MT contacts are plastic

Fewer contacts formed between kinesin and the MT, 170–240 H-bonds and 250–390 nonpolar contacts, of which only 4–10 (H-bond) and 5–16 (nonpolar) had greater than 80% occupancy (*Supplementary file 3*). Majority of nonpolar contacts are by charged or polar residues so that a hydrated interface is maintained (*Figure 3C*). In contrast to intra-kinesin contacts, very few contacts formed or broke irreversibly during the simulation (*Supplementary file 3*). Kinesin-MT contacts can be grouped into the rear (mainly L11 and $\alpha$4), middle (L12 and $\alpha$5), and front ($\beta$5a-L8b, herein called L8/$\beta$5) (*Figure 4C*). The first two interact respectively with $\alpha$ and $\beta$-tubulins, and they are present in all nucleotide states. The front contacts are less robust in the pre-stroke states, lacking high-occupancy nonpolar contacts. This is consistent with the increase of its normal position (*Figure 3A*), higher (weaker) binding energy with $\beta$-tubulin (*Figure 3—figure supplement 1L*), and also with variations in its MT-binding mode in available structures (*Sindelar and Downing, 2007*; *Morikawa et al., 2015*; *Shang et al., 2014*; *Atherton et al., 2014*). Moreover, our analysis agrees with previous alanine-scanning experiments (*Woehlke et al., 1997*). Mutations in L8/$\beta$5 (H156, E157, R161) caused marginal changes in the MT binding affinity, while mutating K252, Y274, and R278 in

the rear and middle parts of the interface affected the MT affinity more strongly, which form high-occupancy contacts in our simulations (*Supplementary file 3*).

Variations in contact occupancy suggest that the kinesin-MT interface is maintained by an ensemble of contacts that do not need to be present simultaneously at any given time. In this way, kinesin may be able to bind to the MT quickly without needing to establish a precise combination of contacts. It also allows a certain degree of mobility of the motor head relative to the MT (*Figure 3A*). Nevertheless, kinesin binds selectively to the cleft on the MT lattice with β-tubulin in front, but not with α-tubulin in front (*Figure 1B*). Although the two tubulins have similar sequence and structure (*Löwe et al., 2001*), we found that some of the residues making contacts with kinesin diverge. Especially, H12 of β-tubulin has several contact residues that are non-homologous to those of α-tubulin (*Figure 4—figure supplement 1C*). Thus, the kinesin-MT interface is tuned so that it permits flexibility in binding, yet it is specific enough to recognize the MT binding site.

## Sw-I hairpin unfolds

Our RMSD and contact analyses show that sw-I is among the most mobile kinesin subdomains. In the ATP-state, although its pseudo-hairpin structure has been suggested to be hydrolysis-competent (*Kull and Endow, 2002*), in all simulations of the post-stroke state, it unfolded (*Figure 2D,E* and *Figure 2—figure supplement 1E*; *Video 1*). The unfolding occurred well after simulation began, at 1.83 $\mu$s (*ATP*), 1.38 $\mu$s (*Kin-only*), and 0.98 $\mu$s (*ADP+Pi*). Even for a 1.73-$\mu$s simulation of an isolated Eg5 in the ATP state (*Parke et al., 2010*), sw-I unfolded at 522 ns (*Video 1*). Prior to full unfolding, contacts within the hairpin partially broke (*Figure 5A,B*), and other contacts within the surrounding domains or with sw-I broke or formed even earlier (*Supplementary file 2*). Thus, unfolding of the sw-I hairpin is a result of gradual changes that accumulate over time, rather than being an isolated event.

After unfolding, $\alpha 3$ at the N-terminal side of sw-I rotated outward, increasing the distance between the two ends of sw-I (*Figure 5A* and *Figure 5—figure supplement 1A*). In the pre-stroke states, $\alpha 3$ generally points outward (larger $\theta_3$ in *Figure 5C*), suggesting a tendency to move outward in the absence of ATP that holds sw-I. To refold into a hairpin, inward rotation of $\alpha 3$ is necessary. Such rotation requires a broader conformational motion of the motor head that may occur over a time scale longer than that of our simulation. The apparent instability of the sw-I hairpin is at odds with its presence in several crystal structures (*Supplementary file 1*). In fact, the hairpin is stabilized by crystal contacts in these structures (*Figure 5—figure supplement 1B–E*). In comparison, the sw-I hairpin in myosin forms extensive contacts with the upper 50 kDa domain (*Figure 5—figure supplement 1F*). However, unfolding of sw-I in our simulation is only partial, where its N-terminal (outer) side separates from the C-terminal side, while the latter maintains contact with ATP. This was also the case for *Kin-only* (*Figure 2E*). In other x-ray structures of kinesin in the ATP (analog) states where sw-I does not adopt a clear hairpin conformation (*Supplementary file 1*), the C-terminal side in contact with the nucleotide is visible, which agrees with the partial unfolding in our simulation (e.g., *Figure 5—figure supplement 1G*).

To further examine the partial unfolding of the sw-I hairpin, we aligned the initial structure of *ATP* and the structure after the hairpin unfolding to high-resolution cryo-EM maps in the ATP states (*Figure 6*). The central β-sheet and $\alpha 4$ were used as alignment references since their conformation varied little during the simulation (*Figure 2A*). To highlight differences between these and the cryo-EM structures, we rigidly docked the structures instead of performing flexible fitting. In both structures with the sw-I hairpin folded and unfolded, the N-terminal side deviates more from cryo-EM maps compared to the C-terminal side. Also, the outward rotation of $\alpha 3$ (~10°; *Figure 5C*) is not enough to show any significant deviation from cryo-EM maps. In *ATP*, the unfolded N-terminal side deviates less from its position in the hairpin state compared to *Kin-only* (*Figure 2D* vs. E). Thus, at cryogenic

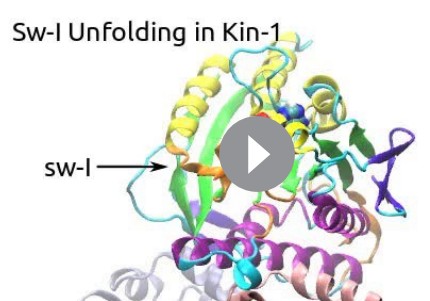

**Video 1.** Unfolding of sw-I in Kin-1 and Kin-5 (Eg5). DOI: https://doi.org/10.7554/eLife.28948.014

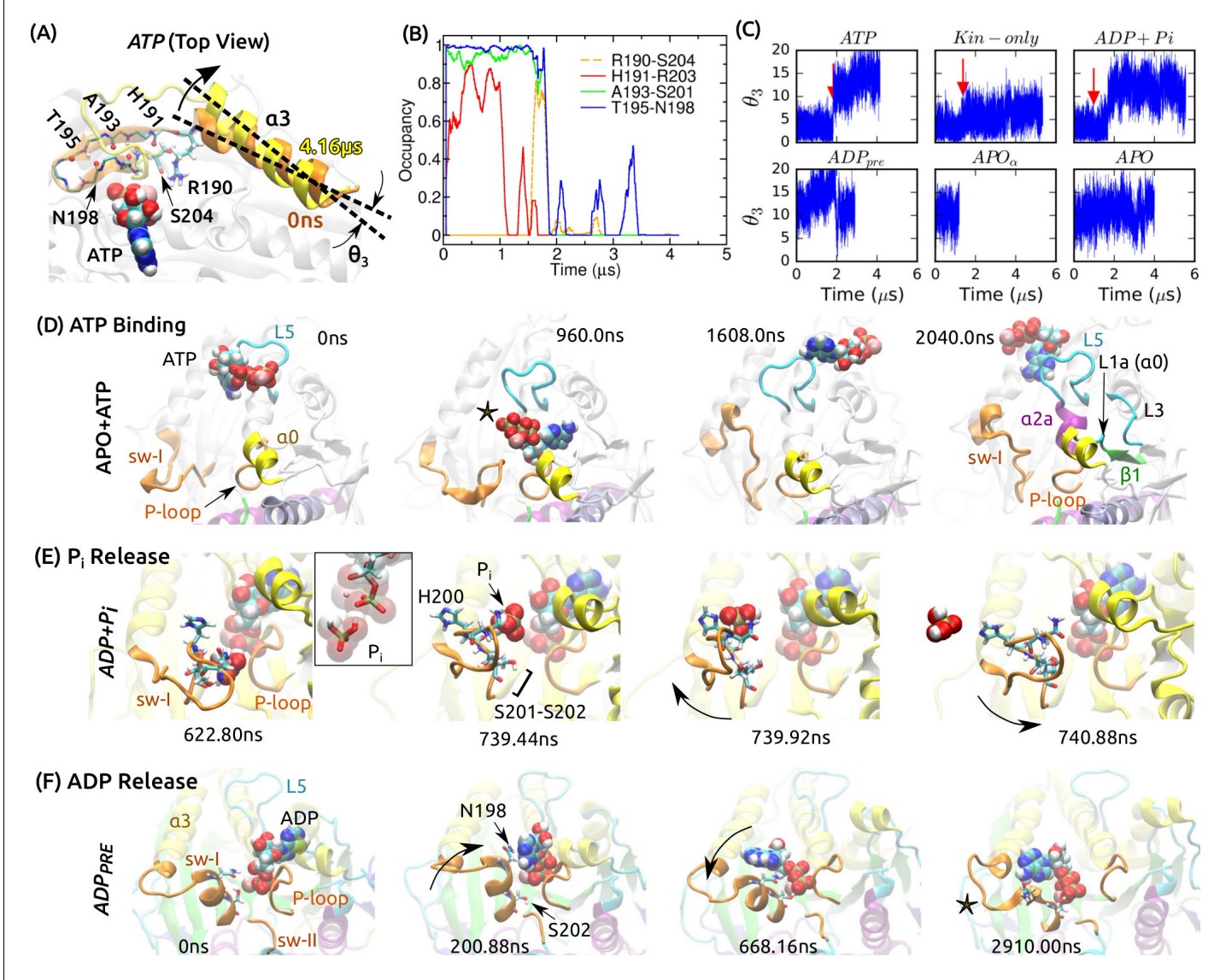

**Figure 5.** Mobility of sw-I in nucleotide processing. (A) Unfolding of the sw-I hairpin in *ATP* (*Video 1* and *Figure 5—figure supplement 1A*). Residues forming backbone H-bonds in the initial hairpin, and the rotation of $\alpha3$ are marked. (B) Occupancy trajectories of contacts for the hairpin in *ATP*. (C) Orientational angle of $\alpha3$ measured relative to the first frame of *ATP*. Red arrow: approximate time at which the sw-I hairpin unfolds. (D) Capturing ATP in the APO state by the '$\alpha0$/L5/sw-I trio' (a 2.04-$\mu s$ simulation; *Video 2*). 960 ns: Adenosine ring is close to its position in the bound state, but the phosphate moiety points outward (star). 1608 ns: Spontaneous formation of an $\alpha$-helical turn in sw-I is visible. 2040 ns: ATP is positioned behind L5. Major domains that made contacts with the moving ATP are labeled (*Figure 5—figure supplement 2A*). (E) $P_i$ release in *ADP+Pi* (*Video 3*; *Figure 5—figure supplement 2B* shows $P_i$ release in Eg5). Box: Magnified view of $P_i$ in contact with ADP. At 739.92 ns, sw-I pulls $P_i$ out (arrow), after which it snaps back (arrow in 740.88 ns). (F) ADP release in *ADP*$_{pre}$ (*Video 4*). Sw-I in an $\alpha$-helical conformation turns and contacts ADP (200.88 ns). Outward rotation of sw-I moves ADP out of P-loop (668.16 ns). Later, sw-I loses its $\alpha$-helical conformation (star in 2910 ns). A magnified view is in *Figure 5—figure supplement 2E*.

DOI: https://doi.org/10.7554/eLife.28948.011

The following figure supplements are available for figure 5:

**Figure supplement 1.** Deformability of the sw-I hairpin.
DOI: https://doi.org/10.7554/eLife.28948.012

**Figure supplement 2.** ATP processing in Kin-1 and Eg5.
DOI: https://doi.org/10.7554/eLife.28948.013

**Figure 6.** Sw-I conformation in cryo-EM structures of the ATP-state kinesin-MT complexes. Yellow/magenta ribbons: Initial structure of *ATP* (PDB 4HNA) and structure at 3.79-$\mu$s that has an unfolded sw-I. Compared to the C-terminal side, the N-terminal side of sw-I aligns less well with cryo-EM maps (arrows), consistent with its mobility in our simulation. References: (A) (*Atherton et al., 2014*), (B) (*Shang et al., 2014*), (C,D) (*Zhang et al., 2015*). Resolutions of the maps are shown in each panel.

DOI: https://doi.org/10.7554/eLife.28948.018

temperatures, the N-terminal side is likely to settle to a hairpin-like state with the C-terminal side as a template, instead of landing in different configurations that lead to low electron density. These findings suggest the mobility of the N-terminal side of sw-I does not contradict existing cryo-EM data.

## Binding of ATP is mediated by the $\alpha_0$/L5/sw-I trio

What would be the functional role of sw-I's mobility? We first consider binding of an ATP molecule to kinesin in the APO state. For the *APO* system, we added a free Mg-ATP and

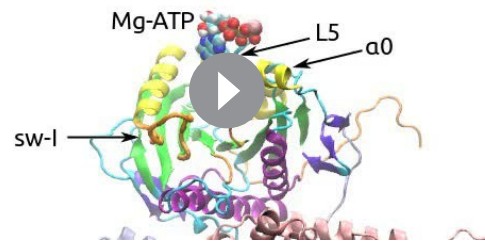

Assisting of ATP binding by trio domains

**Video 2.** Assisting of ATP binding by the α0/L5/sw-I trio. It shows a part of the simulation, demonstrating how an unbound ATP interacts with kinesin.

DOI: https://doi.org/10.7554/eLife.28948.015

performed another 2.04-$\mu$s simulation. To prevent ATP from diffusing away, we imposed a 32 Å radius spherical boundary on ATP around the center of mass of kinesin. During the simulation, ATP formed and broke contacts with various parts of kinesin (*Figure 5D*; *Video 2*). Nonpolar contacts were dominant, with the adenosine ring of ATP pointing toward kinesin and the charged phosphate moiety pointing away (*Figures 5D*, 960 ns, and *Figure 5—figure supplement 2A*). Direct binding of an ATP with the phosphate moiety pointing inward will be unfavorable due to the desolvation penalty for the phosphate moiety and the hydrophobic attraction for the adenosine ring.

ATP binding is more likely a multi-step process orchestrated by the surrounding domains. Among domains whose contact occupancy with ATP was high (labeled in *Figures 5D*, 2040 ns; *Figure 5—figure supplement 2A*), sw-I, $\alpha$0 (including L1a; *Figure 1—figure supplement 1*), and L5 take the shape of a funnel with the P-loop in the middle. During the simulation, the three domains transiently made contacts with ATP or even held it for a while (*Figure 5D*; *Video 2*). This $\alpha_0$/L5/sw-I 'trio' act like an antenna that captures nearby ATP and delivers it to the P-loop. Sw-I, the most mobile member of the trio (higher RMSD than the other two; *Figure 2B*), may be particularly important. When it moves away from the P-loop, it may form contacts with the adenosine ring, so that the phosphate moiety of ATP points towards the P-loop. A closing motion of sw-I will then bring the phosphate moiety in contact with the P-loop. Sw-I's opening and closing motions have been observed in other simulations described below (*cf.*, *Figure 5E,F*). Since ATP is amphiphilic, and since the trio domains closely surround the P-loop, their dynamic role should hold even though a complete binding event was not observed in our simulation – alternative scenarios such as ATP approaching kinesin with the phosphate moiety pointing towards the P-loop, or the trio domains not interacting with the incoming ATP despite their proximity, are physically unlikely.

## Catalytic water molecules are dynamically coordinated

A critical question regarding the mobility of sw-I is whether it can support ATP hydrolysis. As noted above, even when the sw-I hairpin is unfolded, its inner side maintains contact with ATP. In its conserved SSR motif (S201-S202-R203) (*Vale and Milligan, 2000*; *Kull and Endow, 2002*), S201 and S202 contact Mg-ATP with higher than 99% occupancy in both *ATP* and *Kin-only*. Furthermore, R203 contacts E236 of sw-II (*Figure 7A* and *Figure 7—figure supplement 1*). We investigated whether this organization is sufficient to coordinate the catalytic water molecules. A previous *ab initio* calculation on Eg5 suggested a two-water mechanism: The 'lytic' water next to $P_\gamma$ donates an OH group necessary for hydrolysis, and the released H atom travels through the second 'transfer water,' arriving at E236 (*McGrath et al., 2013*). Formation of a two-water bridge between $P_\gamma$ and E236 is thus necessary for hydrolysis.

For *ATP* and *Kin-only*, we calculated the water density map around the phosphate moiety during the last 500 ns. In *ATP*, high-density blobs corresponding to the two catalytic water molecules were found, whereas in *Kin-only*, the density for the transfer water broadened (*Figure 7A,B*). This is because in *Kin-only*, the absence of the support by MT leads to a downward shift of L11 and R203-E236, so that the channel leading to $P_\gamma$ widens (*Figure 7C*). Furthermore, during the simulation period, two-water bridges formed with higher frequency in *ATP* than in *Kin-only* (*Figure 7D*). These suggest that ATP hydrolysis is carried out in a dynamically fluctuating environment where contacts between ATP and sw-I (S201/S202), and between sw-I and sw-II (R203-E236) create a narrow channel that facilitates formation of the two-water bridge between $P_\gamma$ and E236. Our result also explains the higher ATP hydrolysis rate when kinesin is bound to the MT: Without support from $\alpha$-tubulin that keeps L11 ordered, R203-E236 move downward and broaden the channel (*Figure 7C*), thereby reducing the two-water bridge formation. This agrees with the lower but non-zero ATP hydrolysis rate of kinesin in the absence of the MT (*Vale, 1996*).

## Kinesin-bound ATP is torsionally strained

In general, binding of a substrate to an enzyme is believed to induce mechanical strain, thereby lowering the activation energy for cleavage (*Williams, 1993*; *Bustamante et al., 2004*). To check for strain on the kinesin-bound ATP, we measured the internal coordinates of the phosphate moiety in *ATP* and *Kin-only*. For comparison, we performed a 4-ns simulation of an isolated Mg-ATP in water (named *ATP-only*). Between isolated and kinesin-bound ATP, the length of the cleaved $O_\beta$–$P_\gamma$ bond increased from 1.576$\pm$0.032 Å (*ATP*-only; avg$\pm$std) to 1.586$\pm$0.032 Å (*ATP*) and 1.586$\pm$0.033 Å

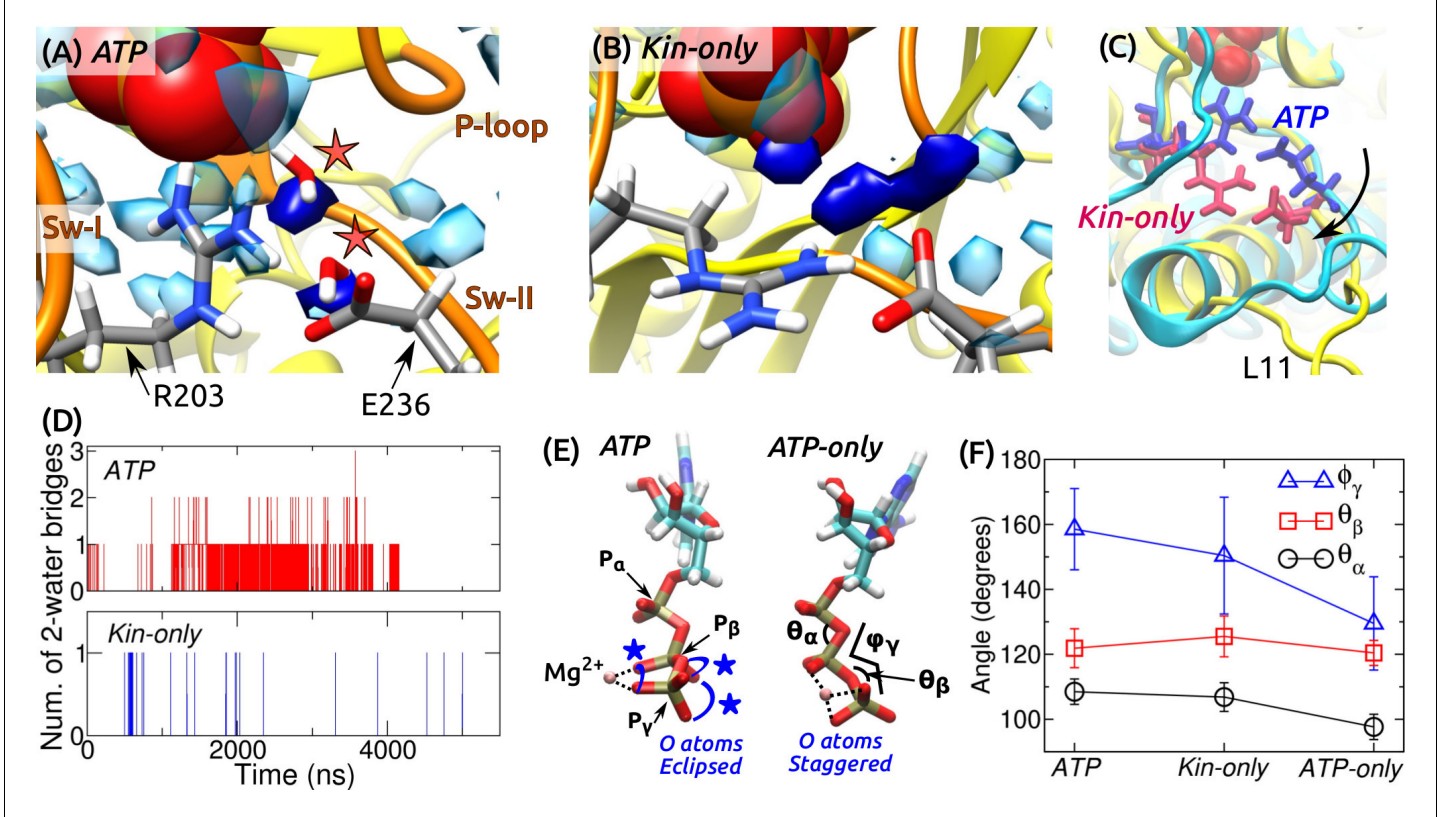

**Figure 7.** ATP hydrolysis mechanism. (**A**) Average water density map near $P_\gamma$ during the last 500 ns of *ATP*. Blobs in semi-transparent blue have water density greater than three times the bulk density (cf., *Figure 3C*). Densities for two catalytic water molecules bridging between $P_\gamma$ and E236 are in dark blue. A coordinate frame where the two-water bridge is present is displayed, with the catalytic water molecules marked by stars. (**B**) Water density map for *Kin-only*. The density close to E236 is broader. (**C**) Alignment of the structures in (**A**) (cyan) and (**B**) (yellow). R203/E236 are in blue (*ATP*) and red (*Kin-only*). In *Kin-only*, downward movement of L11 leads to lowering of R203-E236 (arrow). (**D**) Number of 2-water bridges between $P_\gamma$ and E236 during respective simulations. (**E**) Conformations of Mg-ATP at the end of *ATP* and *ATP-only*. $Mg^{2+}$ forms bidentate (*ATP*) and tridentate (*ATP-only*) contacts with O atoms (dotted lines). O atoms of $P_\beta$ and $P_\gamma$ are eclipsed in *ATP* (blue stars), whereas they are staggered in *ATP-only*. (**F**) Angles defined in (**E**) (avg±std). The dihedral angle $\phi_\gamma$ reflects the eclipsed vs. staggered states.

DOI: https://doi.org/10.7554/eLife.28948.019

The following figure supplement is available for figure 7:

**Figure supplement 1.** Contact occupancy trajectories of R203-ATP (top) and R203-E236 (bottom) in *ATP* and *Kin-only*.
DOI: https://doi.org/10.7554/eLife.28948.020

(*Kin-only*). In the case of the Ras protein, increase in bond length by 0.01 Å has been shown to affect catalysis of GTP (*Klähn et al., 2005*). However, for myosin, although the $O_\beta$–$P_\gamma$ bond elongates in the active site, an energetic analysis reveals no significant destabilization of ATP (*Yang and Cui, 2009*). Furthermore, in the CHARMM param36 force field (*Pavelites et al., 1997*), the equilibrium length of the $O_\beta$–$P_\gamma$ bond is longer, 1.68 Å. Without an *ab initio* calculation of the energetics, it is difficult to assess the impact of stretching the bond on hydrolysis.

Internal angles of ATP had greater changes (*Figure 7E,F*). In particular, the dihedral angle $\phi_\gamma$ increased by 21°–29° when ATP is bound to kinesin. This places the O atoms of $P_\gamma$ in an 'eclipsed' (*cis*) position compared to *ATP-only*, where they are in a more relaxed, 'staggered' (*trans*) position (*Figure 7E*). The different torsional states of $P_\gamma$ also affects contact with $Mg^{2+}$. In *ATP* and *Kin-only*, $Mg^{2+}$ forms bidentate contacts with two O atoms each from $P_\beta$ and $P_\gamma$. In ATP-only, it forms tridentate contacts with two O atoms of $P_\gamma$ and one from $P_\beta$ (*Figure 7E*, dotted lines). Since the increase in the dihedral energy is not substantial (1.2 kcal/mol), torsional angles of the phosphate moiety should readily change when ATP binds to kinesin (calculation using a modified force field that worked well for certain ATP-bound protein structures (*Komuro et al., 2014*), also yielded only

marginal changes in the dihedral energy). But holding only one $O_\gamma$ atom by $Mg^{2+}$ will result in a greater electron withdrawal effect compared to the case when the contact is shared between two oxygens. This permits the lytic water to more easily attack $P_\gamma$ momentarily on the opposite side. Similar dihedral transition and charge redistribution in the phosphate group of GTP upon binding to Ras/GTPase activating protein have been observed (*Rudack et al., 2012*). For myosin, the eclipsing is present in both pre-powerstroke and post-rigor states. The latter state is incapable of hydrolyzing ATP since critical residues in the switch domains are displaced (*Lu et al., 2017*). Although a torsion-based ATPase mechanism may hold across nucleotide triphosphatases, there are likely multiple hydrolysis pathways, whose relative energetics may be determined collectively by the ATP conformation, catalytic water coordination, and conformations of residues immediately surrounding ATP as well as remote domains of the motor (*Lu et al., 2017*).

## Release of hydrolysis products is mediated by the mobile sw-I

*ADP+Pi* models the state after ATP hydrolysis, and $P_i$ was pulled out by sw-I as it moved away from the P-loop (*Figure 5E*; *Video 3*). The gap created between sw-I and ADP was sufficient for $P_i$ to exit above (*Figure 5E*, 739.92 ns). In reality, there may be multiple $P_i$ release paths due to the mobile sw-I. In a similar 2.04-$\mu$s simulation of the Eg5-MT complex, $P_i$ released at 701 ns in a rearward direction (*Figure 5—figure supplement 2B*; *Video 3*).

Recent experiments suggest that the duration of the ADP+$P_i$ state affects the processivity of a kinesin dimer (*Milic et al., 2014*; *Andreasson et al., 2015*; *Mickolajczyk et al., 2015*; *Hancock, 2016*). In the above simulations, $P_i$ was monovalent ($H_2PO_4^-$). In two simulations (3.7 $\mu$s and 3.8 $\mu$s each) of the Eg5-MT complex with a divalent phosphate ($HPO_4^{2-}$; $P_i^{2-}$), $P_i^{2-}$ formed an extensive network of contacts with Mg-ADP and sw-I, and did not release (*Figure 5—figure supplement 2C,D*). $P_i^{2-}$ is a high-energy transition state where the proton released after hydrolysis is added to convert it to $P_i$ (*McGrath et al., 2013*). Since the proton can instead release into bulk water, the time of conversion from divalent to monovalent phosphate may depend on the time scale of proton transfer and other factors such as conformational fluctuation of kinesin. The phosphate release time also depends on the orientation of the phosphate in the nucleotide pocket. In another 2-$\mu$s simulation of Kin-1 with a monovalent $P_i$ (as in *ADP+Pi*), its lone oxygen atom formed a contact with $Mg^{2+}$, and release did not happen until the end of the simulation, analogous to the situation in *Figure 5—figure supplement 2D*. While various factors affect the time scale of $P_i$ release, since sw-I firmly contacts $P_i$ and is mobile, its outward motion is expected to be involved in driving the release.

Sw-I also facilitates ADP release, which was observed in *ADP*$_{pre}$ (*Figure 5F*). At 201 ns, it swung toward ADP and its conserved N198 formed nonpolar contact with the adenosine ring (*Figure 5F*). This contact was transient and broke again. After a number of attempts, ADP was gradually pulled out (*Figure 5—figure supplement 2E*; *Video 4*). Sw-I then lost its $\alpha$-helical conformation (*Figures 5F*, 2910 ns). It is unclear whether the $\alpha$-helical state of sw-I is required for ADP release. A possible advantage is that the helix is more rigid than a disordered state, and it can exert a lever action for pulling ADP out of the P-loop. In the simulation of ATP binding, sw-I spontaneously formed an $\alpha$-helical turn (*Figures 5D*, 2040 ns), which indicates that it can transition between disordered and helical states unless the SSR motif is

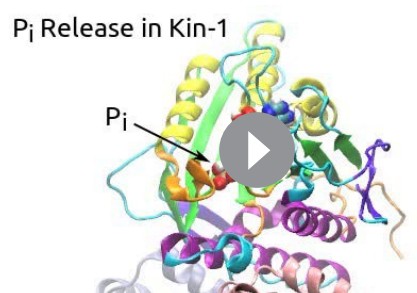

**Video 3.** P release in Kin-1 and Kin-5 (Eg5).
DOI: https://doi.org/10.7554/eLife.28948.016

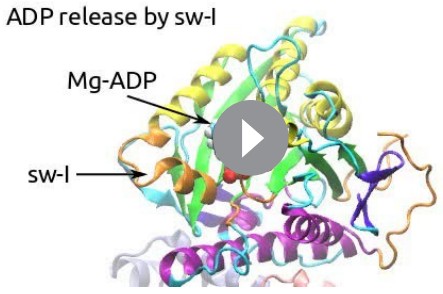

**Video 4.** Process of ADP release by sw-I. Only a part is shown.
DOI: https://doi.org/10.7554/eLife.28948.017

stabilized by a bound ATP.

## Discussion

The present results and previous findings lead us to propose a detailed model of kinesin dimer motility in which subdomain dynamics plays an essential role (*Figure 8* and *Video 5*). It begins with the hydrolysis of the bound ATP in the rear head, which involves the sw-I–II connection, dynamic water coordination, and torsional strain in catalyzing ATP hydrolysis (*Figure 8A*). After ATP hydrolysis, the rear head changes its position and orientation slightly, which allows the front head to release ADP and fully bind to the MT (*Figure 8B*). Until the rear head releases $P_i$ and detaches, ATP binding to the front head is prevented ('gated'; *Figure 8C*). Once the rear head unbinds, possibly coupled with unfolding of $\alpha 4$ (see below), ATP binding to the front head occurs, assisted by the $\alpha 0$/L5/sw-I trio domains (*Figure 8D*). The resulting formation of the CNB in the front head generates the power stroke in which the rear head is thrown forward and begins a diffusive search for the next binding site. The E-hook of the MT helps with capturing what is now the front head (*Figure 8E*).

The present work focuses on the properties of a single kinesin head that is likely to form the basis for diverse motility characteristics of different kinesins. In this regard, while the above model of dimer motility describes how it may be achieved by subdomain dynamics, the present model cannot address aspects pertaining specifically to the dimer motility, which would require knowledge of the communication between the two heads. Nevertheless, several general conclusions can be made. Our model highlights the active nature of nucleotide processing, where binding of ATP, hydrolysis, and release of hydrolysis products are mediated by concerted motions of mobile subdomains. The inherent mobility of sw-I is consistent with an experiment based on fluorescence resonance energy transfer (*Muretta et al., 2015*). It was shown in the paper that sw-I in both Kin-1 and Kin-5 bound to the MT stayed in an 'open' state more than 50% (mole fraction), even in the ATP state, as opposed to the 'closed' state that was assumed to take the hairpin conformation. The higher mobility or deformation of sw-I for an isolated kinesin compared to the MT-bound kinesin with ATP (*Figure 2D, E*) is also consistent with a previous electron paramagnetic resonance experiment (*Naber et al., 2003*). In the ATP state, the mobility of sw-I is limited such that its C-terminal side maintains contacts with ATP and sw-II that are necessary to support the hydrolysis. This was the case even for an isolated kinesin, whose lower ATP hydrolysis rate is likely due to the widening of the nucleotide pocket that reduces coordination of the catalytic water molecules (*Figure 7*).

The conformational fluctuation of the outer (N-terminal) side of sw-I may allosterically affect the catalytic water coordination and the precise orientation of residues of the SSR motif on the C-terminal side. In the case of myosin, these factors have been shown to affect the energetics of ATP hydrolysis (*Lu et al., 2017*). The hairpin conformation of sw-I could be more advantageous for hydrolysis than the unfolded state. However, since the hydrolysis reaction itself proceeds over a picosecond time scale (*McGrath et al., 2013*), the hairpin does not need to stay stably folded. Thus, variations in the sequence of the distal part of sw-I may provide an allosteric mechanism to fine-tune ATP hydrolysis and other kinetic rates by controlling its flexibility. This idea is supported by the behavior of the R190A/D231A mutant (*Cao et al., 2014*), which is expected to destabilize or prevent the hairpin state (*Figure 5—figure supplement 1A*). Rather than abolishing hydrolysis, its catalytic rate is 22% of the wild-type value. This demonstrates that the hairpin state may promote hydrolysis, but it is not required. Another illuminating feature of the dynamic role of sw-I is that the ATP hydrolysis rate depends on whether kinesin is bound to the MT filament or to unpolymerized tubulin (*Alonso et al., 2007*; *Gigant et al., 2013*). Since sw-I is the most labile element within the nucleotide pocket, its conformational motion may be affected the most when kinesin is bound to an unpolymerized tubulin, so as to influence the catalytic rate.

We propose that $\alpha 0$ is the structural element responsible for ATP gating in the front head (*Figure 8C*). For the gating, the rearward orientation of the NL, rather than its tension, is essential (*Clancy et al., 2011*; *Andreasson et al., 2015*). In x-ray structures of kinesins with a rearward-docked NL, it interacts with the 3-stranded $\beta 1$ domain, which is linked to the C-terminal end of $\alpha 0$ (*Figure 1—figure supplement 1*) (*Sablin and Fletterick, 2004*; *Guan et al., 2017*). This raises the possibility that the positional fluctuation of $\alpha 0$ (*Figure 2*) is controlled by the interaction between the NL and $\beta 1$, thereby affecting the ATP binding. Further tests are needed to validate this proposal.

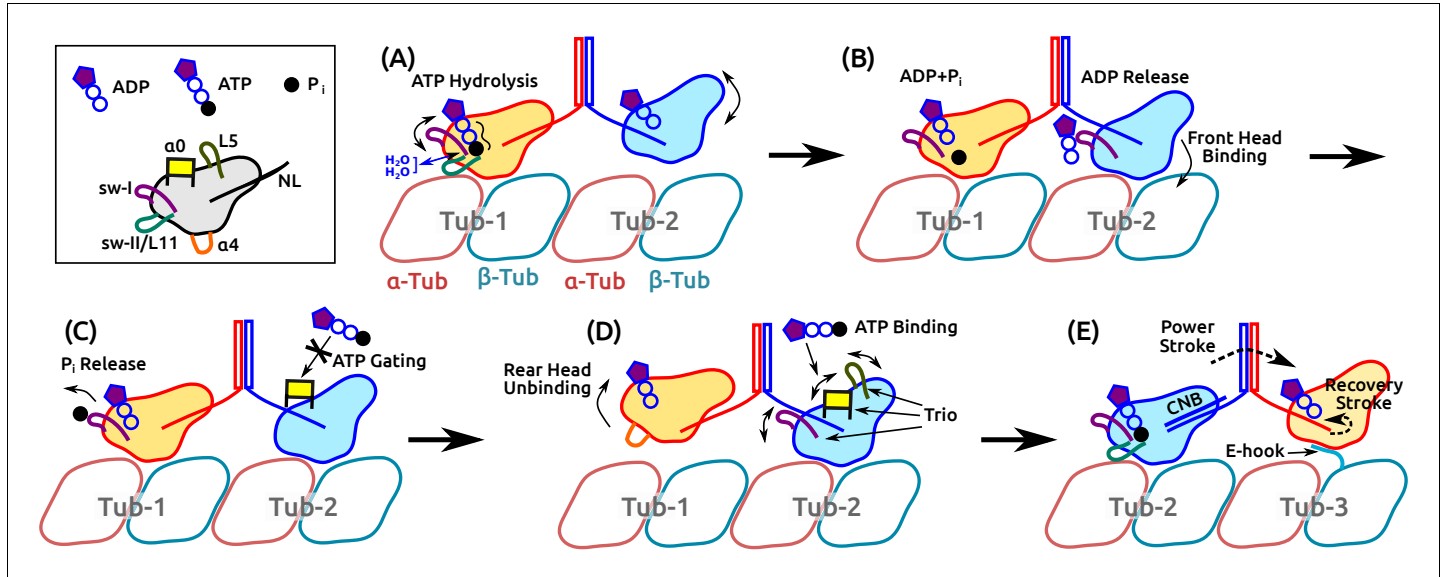

**Figure 8.** Motility of a Kin-1 dimer driven by subdomain dynamics. *Video 5* shows an overview of the process. In each panel, only relevant domains are shown. Tubulin dimers are labeled Tub-1–Tub-3. (**A**) ATP hydrolysis is driven by sw-I motion, sw-I–II connection, dynamic water coordination, and torsional strain (wavy line) on $P_\gamma$. The front head may not be fully bound. (**B**) ATP hydrolysis in the rear head allows full binding of the front head to Tub-2 (*Milic et al., 2014*), which releases ADP, mediated by sw-I. (**C**) Until $P_i$ releases from the rear head (assisted by sw-I), ATP binding to the front head is gated (*Andreasson et al., 2015*; *Hancock, 2016*), potentially by $\alpha 0$ that is close to the rearward-pointing NL. (**D**) Rear head (ADP state) unbinds, possibly coupled with unfolding of $\alpha 4$. This allows ATP binding to the front head, assisted by the trio domains. (**E**) Power stroke is generated by the CNB formation (*Hwang et al., 2008*). The new front head contacts Tub-3 via interaction with the E-hook (*Sirajuddin et al., 2014*).
DOI: https://doi.org/10.7554/eLife.28948.021

The relatively low specificity of the hydrated kinesin-MT interface (*Figure 3C*, *Figure 4C*, *Supplementary file 3*) is suited for rapid interaction with MTs in vivo (*Leduc et al., 2012*). In *ADP +Pi*, we did not observe unbinding of the motor head after $P_i$ release, as is expected for the ADP-state kinesin. It is significant that *ADP+Pi* had more high-occupancy contacts with the MT than the other states (*Supplementary file 3*; this was the case even when only the last 500 ns, well after $P_i$ release in *ADP+Pi*, were considered), and its MT-binding energy was among the strongest (*Figure 3—figure supplement 1L*). It appears to us unlikely that extension of the simulation time will lead to detachment of the motor head. To disrupt the kinesin-MT interface, we suggest that a conformational transition must occur. A likely subdomain for this transition is the N-terminus of $\alpha 4$. It unfolded transiently in *ADP+Pi* and more extensively in *Kin-only* (*Figure 2E*). Conformational fluctuation of the unfolded $\alpha 4$ can disrupt the rear part of the kinesin-MT contact (*Figure 4C*), leading to detachment. This picture agrees with the recent finding that $P_i$ release is required for unbinding of kinesin from the MT (*Milic et al., 2014*). The presence of $P_i$ in the nucleotide pocket suppresses the unfolding of $\alpha 4$, thereby keeping the motor head bound to the MT. This is an interesting subject for studies beyond the present simulations.

Compared to a static mechanism that requires a specific structure, the dynamic mechanism for nucleotide processing and MT binding found here, provides greater flexibility in fine-tuning time scales and affinities. For example, the conserved residues that contact ATP cannot account for differences in catalytic rates among various kinesins. Altering residues that do not directly contact ATP can affect the

**Motility Cycle of Kinesin-1
Driven by Subdomain Dynamics**

Wonmuk Hwang
Matthew J. Lang
Martin Karplus

**Video 5.** Illustrative model of the motility of a kinesin dimer.
DOI: https://doi.org/10.7554/eLife.28948.022

conformational dynamics, which could in turn influence the frequency of 2-water bridge formation and charge fluctuation around $P_\gamma$, thereby controlling the catalytic rate.

Of interest are experimental means to test the dynamic roles of subdomains by introducing mutations. Since the N-terminal side of sw-I does not contact ATP directly, it may be possible to mutate it to alter the dynamical properties of sw-I without severely impairing motility. For example, a more flexible sw-I may enhance rates for all phases of nucleotide processing. However, caution must be exercised here, since if sw-I is made too flexible, it may disrupt contacts with the nucleotide. Similarly, elongating $\alpha 0$ by lengthening L1a and L1b (*Figure 1—figure supplement 1E*) may increase the ATP binding rate, but it may also affect the gating behavior (*Figure 8C*). The design of mutants and their predicted behavior will require careful analysis and simulations.

We also showed that the elastic energy of the curvature of the central $\beta$-sheet or deformation of the MT, are unlikely to drive motility (*Figure 2* and *Figure 2—figure supplement 1*). The aspect ratio of the motor head is too small to store any significant deformational energy. Even for myosin V that has a larger aspect ratio, there is little evidence that the twist of its $\beta$-sheet has any strong energetic role (*Cecchini et al., 2008*). Furthermore, the hydrated and dynamic kinesin-MT interface is unlikely to induce substantial strain in the motor head. Instead, small amount of strain may play a role in fine-tuning kinetic rates.

The free energy change upon NL docking in the absence of load (0.7–2.9 kcal/mol) (*Rice et al., 2003*; *Muretta et al., 2015*) is much smaller than the maximum work done near the stall force (5.8–8.1 kcal/mol). Additional energy is likely to originate from the CNB formation (*Hwang et al., 2008*; *Khalil et al., 2008*) and binding of the front head to the MT (*Figure 8E*). Since the hydrolysis energy of ATP thermalizes rapidly (on the picosecond time scale), it is unlikely to have any direct role. However, the differential binding energy of ATP and its hydrolysis products are likely to be important in triggering the large transitions of the motility cycle. Thus, while the net free energy change after a motility cycle may be that of hydrolyzing an ATP, there is a large free energy flow between kinesin and the environment during each phase of the cycle. In this regard, kinesin's mobile subdomains are 'free energy transducers.'

The present results provide the basis for understanding the role of local subdomain dynamics in the kinesin motility cycle. We emphasize that the extension of the simulations to multiple microseconds for each state, made possible by the use of Anton, played an important role in obtaining converged results. Suggestions are made concerning elements that will have to be tested by future experiments and simulations. This is important because the variation in structural rigidity, hydration, and protein-protein interaction found in the simulations provide a dynamic description of how kinesin works that is significantly different from conclusions based solely on static crystal and cryo-EM structures. Given the commonality among translocating motor proteins (*Hwang and Lang, 2009*), it is likely that local subdomain dynamics plays active roles for driving conformational changes and reactions, more generally.

# Materials and methods

## PDB structures used

1. *ATP*: PDB 4HNA (3.19 Å resolution). Kinesin-MT complex with an ATP analogue (*Gigant et al., 2013*).
2. *Kin-only*: PDB 4HNA without the MT.
3. *ADP+Pi*: The coordinate frame of *ATP* at 1043 ns, with ATP converted to ADP and $P_i$.
4. *ADP$_{pre}$*: PDB 2P4N (9 Å resolution), a cryo-EM structure of nucleotide-free kinesin-MT complex (*Sindelar and Downing, 2007*). The fitted kinesin x-ray structure was based on PDB 1BG2 (1.8 Å resolution), which has a bound ADP and sw-I in $\alpha$-helical conformation (*Kull et al., 1996*). The missing L11 and the N-terminal part of $\alpha 4$ in the MT-facing domain were modeled after PDB 4HNA.
5. *APO$_\alpha$*: Same as *ADP$_{pre}$*, with ADP removed. Sw-I was left in the $\alpha$-helical conformation.
6. *APO*: PDB 4LNU (2.19 Å resolution). Nucleotide-free kinesin-MT complex (*Cao et al., 2014*).

Among the above, PDB 4HNA and 4LNU are x-ray structures of kinesin-MT complexes respectively in ATP (ATP analogue) and APO states. Although the tubulin dimers in these structures are slightly

curved, it has been shown not to affect the kinesin-MT interface (*Gigant et al., 2013*; *Cao et al., 2014*). We thus did not straighten the tubulin structure.

For Kin-5, we used the following structures:

1. PDB 4AQV (9.70 Å resolution): Cryo-EM structure of Eg5 bound to the MT in the ATP-state (*Goulet et al., 2012*). This corresponds to *ATP*.
2. PDB 3HQD (2.19 Å resolution): Motor head of Eg5 (no MT) in the post-stroke ATP-state (ATP analog) (*Parke et al., 2010*). This corresponds to *Kin-only*.

## System preparation

We constructed the kinesin structure up to the NL (M1–A337), excluding the $\alpha$-helical stalk. The C-terminal end of a tubulin has 13-aa glutamate-rich E-hook that are invisible in x-ray structures due to its flexibility. For $\alpha$ and $\beta$ tubulins, we omitted the last 9 (E443–Y451) and 4 (E452–A455) residues of E-hooks, respectively. These truncations render the system size to fit within Anton. The E-hook of $\alpha$ tubulin is located on the minus end side of a tubulin dimer (at the left end of $\alpha$H12 in *Figure 1C*) and is away from the kinesin motor head. The E-hook of $\beta$ tubulin locates on the right side of the motor head (at the end of $\beta$H12 in *Figure 1C*). Being negatively charged and flexible, E-hooks are known not to affect kinesin in the MT-bound state, and it is more important for making non-specific electrostatic contacts with an unbound head (*Figure 8E*) (*Lakämper and Meyhöfer, 2005*; *Sirajuddin et al., 2014*). Truncations of E-hooks are thus unlikely to affect our result for MT-bound kinesins.

For each system, the protein structure was placed in a cubic TIP3P water box of linear size 113–119 Å (for kinesin-MT complex; 88 Å for *Kin-only*) and it was made electrically neutral by adding ions to about 50 mM concentration. The number of atoms in our systems were in the 150,000–170,000 range (65,000 for *Kin-only*). A periodic boundary condition was applied to the box.

## Preparatory simulation

In preparation for simulations on Anton, the solvated system was simulated using CHARMM (*Brooks et al., 2009*) on a conventional computer cluster. Initially, a series of energy minimization procedure was done with harmonic constraints applied to proteins and nucleotides, which were gradually reduced to zero in successive 200-step minimization cycles. Next, the system underwent heating (100 ps) and equilibration (200 ps) runs under 1-atm pressure. During heating to 300 K, harmonic constraints were applied to backbone heavy atoms of proteins except for the 4-aa N-terminal end of kinesin's CS and the C-terminal E-hook domains of MT that are flexible. The spring constant of the harmonic constraint was 1 kcal/mol·Å$^2$ during heating, and 0.5 kcal/mol·Å$^2$ during equilibration. It was further reduced to 0.25 kcal/mol·Å$^2$, with only $C_\alpha$ atoms restrained (excluding those of the flexible domains noted above), and simulation continued for 2 ns using the constant temperature (300 K) and pressure (1 atm) (CPT) dynamics method implemented in CHARMM. The final phase of the preparatory run lasted 2 ns with 0.5-kcal/mol·Å$^2$ harmonic constraints applied to $C_\alpha$ atoms of the loops of tubulins that are near the interface with neighboring MT protofilaments (aa 57–61, 83–88, and 279–286, for both tubulins). They are located on the bottom in *Figure 1B*, and restraining them mimics the effect of the tubulin dimer embedded within a polymerized MT. For *Kin-only* that lacks the MT, we harmonically restrained the $C_\alpha$ atoms of L229–D231 of $\beta$7 (*Figure 1—figure supplement 1*) with a 0.1-kcal/mol·Å$^2$ spring constant. These atoms are located approximately at the center of mass of the motor head, and the weak restraint suppresses translational diffusion of kinesin.

For simulation, the CHARMM param36 force field was used. For *ADP+Pi*, the monovalent form of $P_i$ ($H_2PO_4^-$) was constructed based on phosphate parameters in the param22 force field. The SHAKE algorithm was applied to fix the length between hydrogen and its base heavy atom. The integration time step was 2 fs.

## Simulation on Anton

We wrote a Python script to convert the CHARMM restart file at the end of the preparatory run to the the Desmond Maestro format file, which was further processed using the Anton software. The CHARMM param36 force field was used through the Viparr utility of Anton. Harmonic restraints of spring constant 0.25 kcal/mol·Å$^2$ were applied to $C_\alpha$ atoms of the same residues of the MT loops as

in the last phase of the preparatory simulation. SHAKE was applied to hydrogen atoms, with a 2-fs integration time step. The multigrator integration method of Anton was used under a CPT (300 K, 1 atm) condition. Coordinates were saved every 0.24 ns. After simulation, coordinate trajectories were converted to CHARMM DCD format files by using VMD (*Humphrey et al., 1996*), for analysis using CHARMM. All simulations were carried out on Anton (*Shaw et al., 2009*) except for *APO*, which was on the newer Anton-2 machine (*Shaw et al., 2014*).

## Curvature of the central $\beta$-sheet of kinesin

We considered seven strands within the central $\beta$-sheet of kinesin: $\beta 1$ (V11–F15), $\beta 3$ (T80–G85), $\beta 4$ (I130–Y138), $\beta 5$ (I142–D144), $\beta 6$ (S206–K213), $\beta 7$ (K226–L232), and $\beta 8$ (T296–C302) (*Figure 1—figure supplement 1*). This choice excludes regions of $\beta 4$, $\beta 6$, and $\beta 7$ at the front end of the motor head that deformed in some simulations (*Figure 2G,H* and *Figure 2—figure supplement 1G–J*). To calculate curvature, the central $\beta$-sheet in each coordinate frame was oriented to a reference kinesin structure whose least-square-fit plane was oriented to the $xy$-plane of the Cartesian coordinate system and the center of mass positioned at the coordinate origin. In this configuration, $z$-coordinates of C$\alpha$ atoms within the central $\beta$-sheet were parameterized by their $x$ and $y$ coordinates and were fit using the quadratic expansion (*Sun et al., 2003*)

$$z(x,y) = a_0 + a_1 x + a_2 x^2 + a_3 y + a_4 xy + a_5 y^2, \tag{1}$$

where $\{a_i\}$ ($i = 0 \text{ to } 5$) are fitting parameters that vary among coordinate frames. Fitting was done using the SciPy package of Python. Examples of fitting surfaces are in *Figure 2I*. For a given frame, the mean and Gaussian curvatures are given by $M = 2(a_2 + a_5)$ and $G = a_4^2 - 4a_2 a_5$ (*Sun et al., 2003*).

To calculate the potential of mean force (PMF) versus the curvature, we calculated a 2-dimensional histogram of $M^2$ and $G$ normalized by the maximum count, $\rho(M^2, G)$. PMF is given by $-k_B T \ln \rho$ (*Figure 2—figure supplement 1K*). To align PMFs for *ATP* and *APO* (*Figure 2J*), we identified bins of the histogram that have nonzero counts in both simulations. We determined the constant free energy shift $\Delta$ that needs to be added to the PMF for *APO* so that the mean-square difference of the two PMFs in the overlapping bins is minimized. The mean-square difference was calculated weighted by the histogram counts of respective PMFs. Denote the histogram values of *ATP* and *APO* in the $k$-th bin within the overlap region by $\rho_k^{\text{ATP}}$ and $\rho_k^{\text{APO}}$, respectively, and similarly denote their PMFs by $E_k^{\text{ATP}}$ and $E_k^{\text{APO}}$. Minimizing $\sum_k \rho_k^{\text{ATP}} \rho_k^{\text{APO}} (E_k^{\text{ATP}} - E_k^{\text{APO}} - \Delta)^2$ yields

$$\Delta = \frac{\sum_k \rho_k^{\text{ATP}} \rho_k^{\text{APO}} (E_k^{\text{ATP}} - E_k^{\text{APO}})}{\sum_k \rho_k^{\text{ATP}} \rho_k^{\text{APO}}} \tag{2}$$

where the sum is for bins in the overlap region. We added $\Delta$ to the PMF for *APO*, and the merged PMF $E_k$ for the $k$-th bin in the overlap region was set to

$$E_k = \frac{\rho_k^{\text{ATP}} E_k^{\text{ATP}} + \rho_k^{\text{APO}} (E_k^{\text{APO}} + \Delta)}{\rho_k^{\text{ATP}} + \rho_k^{\text{APO}}}. \tag{3}$$

## Measuring motor head motion relative to the microtubule

To calculate the position and orientation of kinesin relative to the microtubule (*Figure 3A,B*, *Figure 3—figure supplement 1A–G*), we used the C$_\alpha$ atoms of the following domains:

- Central $\beta$-sheet used for the curvature calculation.
- $\alpha 4$: I254–V264.
- $\alpha 6$: S310–Q320.
- $\beta 5a/b$: V155–E157 ($\beta 5a$) and Y164–K166 ($\beta 5b$).

Each coordinate frame was aligned to the first frame of *ATP*, with the C$_\alpha$ atoms of H12 helices (aa417–432) of $\alpha$- and $\beta$-tubulins as reference for alignment. Let $\boldsymbol{R}_{\alpha-tub}$, and $\boldsymbol{R}_{\beta-tub}$ be the centers of masses of H12 in respective tubulins. An orthonormal triad $\{\boldsymbol{u}_L, \boldsymbol{u}_N, \boldsymbol{u}_T\}$ was constructed in the following way (*Figure 3—figure supplement 1A*):

- Longitudinal direction: $\boldsymbol{u}_L \propto (\boldsymbol{R}_{\beta-tub} - \boldsymbol{R}_{\alpha-tub})$.

- Normal direction: Let $\boldsymbol{u}_{\alpha 4}$ be the axis vector of $\alpha 4$ in the reference structure (first frame of *ATP*) pointing to the right, from the N- to C-termini of $\alpha 4$. The unit vector in the normal direction was set as $\boldsymbol{u}_N \propto (\boldsymbol{u}_{\alpha 4} \times \boldsymbol{u}_L)$.
- Transverse direction: $\boldsymbol{u}_T = \boldsymbol{u}_L \times \boldsymbol{u}_N$.

Let $r_\beta$ be the center of mass of the central $\beta$-sheet. We projected $(r_\beta - \boldsymbol{R}_{\alpha-\mathrm{tub}})$ onto the three directions of the triad. Differences of these projections from those of the reference structure were defined respectively as the longitudinal ($\Delta_L$), normal ($\Delta_N$), and transverse ($\Delta_T$) displacements. Displacements of $\alpha 4$, Q320, and $\beta 5a/b$ were measured similarly.

To calculate the orientation of the motor head, we used an approximately rectangular section of $\beta 4$ (I130–E136), $\beta 6$ (S206–V212), and $\beta 7$ (K226–D231). Using their $C_\alpha$ atoms, we calculated the major and normal axes of the least-square-fit plane, $\boldsymbol{v}_{\beta 1}$ and $\boldsymbol{v}_{\beta 2}$, respectively. We projected $\boldsymbol{v}_{\beta 1}$ onto the plane spanned by $\boldsymbol{u}_L$ and $\boldsymbol{u}_N$, and measured the forward tilt angle $\theta_\beta$ as the angle between this projection and $\boldsymbol{u}_L$. The azimuthal angle $\phi_\beta$ was measured as the angle between the projection of $\boldsymbol{v}_{\beta 1}$ onto the plane spanned by $\boldsymbol{u}_L$ and $\boldsymbol{u}_T$, with $\boldsymbol{u}_L$. The transverse tilt angle $\omega_\beta$ was between the projection of $v_{\beta 2}$ onto the plane spanned by $\boldsymbol{u}_N$ and $\boldsymbol{u}_T$, with $\boldsymbol{u}_N$. Increase in $\phi_\beta$ is associated with clockwise rotation of the motor head when viewed from top, and for $\omega_\beta$, it is counterclockwise rotation when viewed from the MT plus end. Since the central $\beta$-sheet is tilted to the left, $\omega_\beta$ is typically negative, and in *Figure 3—figure supplement 1F*, $-\omega_\beta$ was plotted. A larger (less negative) $\omega_\beta$ indicates that the motor head tilts more to the right.

Azimuthal ($\phi_\alpha$) and transverse tilt ($\omega_{\alpha 6}$) angles of $\alpha 6$ were similarly measured using the projection of the axis of $\alpha 6$ on respective planes. The orientation angle $\psi_{\alpha 4}$ of $\alpha 4$ was measured between its axis and $\boldsymbol{u}_T$.

## Hydration analysis

To calculate the water density map for the kinesin-MT interface (*Figure 3C*), we adopted a method that we developed previously (*Ravikumar and Hwang, 2011*). Coordinate frames were aligned to the first frame of *ATP* with $C_\alpha$ atoms of the reference domains consisting of $\alpha 4$ (E250–E270), and parts of H11–H12 of $\alpha$-tubulin (F395–E420) and H12 of $\beta$-tubulin (M425–Y435). A search box was set whose boundary is at least 15 Å away from any atom in the above domains. The box was divided into a cubic grid of linear size 0.7 Å. For each cell in the grid, the fraction of frames where a water oxygen is found was calculated and divided by the volume of the cell ($0.7^3$ Å$^3$). The map was saved into an MRC electron density map format file and visualized using UCSF Chimera (*Pettersen et al., 2004*). The water density map around the phosphate moiety of ATP (*Figure 7A,B*) was calculated similarly, with $C_\alpha$ atoms of the P-loop (G85–G90) and $P_\gamma$ as positional reference for aligning coordinate frames.

To calculate the buried area in the kinesin-MT interface (*Figure 3D*), for each coordinate frame, we used a 1.4 Å probe radius to calculate the solvent accessible surface area of kinesin and the MT, together and separately, and measured their difference.

## Binding energy calculation

Calculation of the binding energy was done based on a previously developed method (*Zoete et al., 2005*). Briefly, for each coordinate frame during the last 500 ns, the following energies were measured: van der Waals (Lennard-Jones), electrostatic, generalized Born solvation free energy, and the nonpolar energy. Calculation of energy terms was done using the Generalized Born with a simple SWitcthing (GBSW) module of CHARMM (*Im et al., 2003*). To find the binding energy, we used the kinesin motor head, $\alpha$-tubulin, and $\beta$-tubulin, individually or in combination. For example, between the motor head and the $\alpha$-tubulin, we measured $E_K$ for kinesin, $E_\alpha$ for the $\alpha$-tubulin, and $E_{K\alpha}$ for them together ($E$ denotes an energy term). Their binding energy was defined as $E_{K\alpha} - E_K - E_\alpha$. Similar calculations were done for the binding energy between kinesin and the $\beta$-tubulin, and kinesin and the whole tubulin dimer.

## Contact analysis

For each coordinate frame in a simulation, hydrogen bonds (H-bonds) were identified with the donor-acceptor distance cutoff of 2.4 Å. A residue pair was considered to form a nonpolar contact if the pair has neutral atoms (partial charge less than 0.3$e$; $e=1.6 \times 10^{-19}$ C) that are closer than 3.0 Å.

The occupancy of a bond is the fraction of frames over which the bond is formed during the simulation, and its occupancy trajectory is the rolling (running) average with a 96-ns (400 frames) window.

To identify formation and breakage of a bond during simulation, we calculated its occupancy for the first and last 200 frames (48 ns), respectively. If the initial occupancy is less than 0.05, and the final one is greater than 0.5, the bond was regarded to have formed during the simulation, and *vice versa* for identifying bonds that broke. We traced the trajectory forward (bond formation) or backward (bond breakage) in time and located the time point where the local occupancy became greater than 0.5, as the corresponding transition time (*Figure 4—figure supplement 1A*).

## Acknowledgements

This work was supported in part by the National Institutes of Health (NIH) grant R01GM087677. The work done at Harvard University was supported by the CHARMM Development Project. Anton/Anton-2 computer time was provided by the Pittsburgh Supercomputing Center (PSC) through the NIH grant R01GM116961. Anton at PSC was generously made available by DE Shaw Research. We also used machines at the Texas A&M Supercomputing Facility and Texas Advanced Computing Center at UT Austin. We thank the reviewers for their helpful comments. The open access publishing fees for this article have been covered by the Texas A&M University Open Access to Knowledge Fund (OAKFund), supported by the University Libraries and the Office of the Vice President for Research.

## Additional information

### Funding

| Funder | Grant reference number | Author |
| --- | --- | --- |
| National Institutes of Health | R01GM087677 | Wonmuk Hwang<br>Matthew J Lang |
| Pittsburgh Supercomputing Center | Anton Supercomputer | Wonmuk Hwang<br>Martin Karplus |
| Texas Advanced Computing Center | | Wonmuk Hwang |
| Texas A&M Supercomputing Facility | | Wonmuk Hwang |
| Texas A and M University | Open Access to Knowledge Fund | Wonmuk Hwang |
| CHARMM Development Project | | Martin Karplus |

The funders had no role in study design, data collection and interpretation, or the decision to submit the work for publication.

### Author contributions

Wonmuk Hwang, Conceptualization, Software, Formal analysis, Funding acquisition, Validation, Investigation, Visualization, Methodology, Writing—original draft, Writing—review and editing; Matthew J Lang, Martin Karplus, Conceptualization, Funding acquisition, Writing—original draft, Writing—review and editing

### Author ORCIDs

Wonmuk Hwang  http://orcid.org/0000-0001-7514-3186

### Decision letter and Author response

Decision letter https://doi.org/10.7554/eLife.28948.029
Author response https://doi.org/10.7554/eLife.28948.030

# Additional files

**Supplementary files**

• Supplementary file 1. Conformational variability of kinesin

DOI: https://doi.org/10.7554/eLife.28948.023

• Supplementary file 2. Intra-kinesin hydrogen bonds (HB) and nonpolar (NP) contacts that break or form during each simulation

DOI: https://doi.org/10.7554/eLife.28948.024

• Supplementary file 3. Kinesin-MT contacts.

DOI: https://doi.org/10.7554/eLife.28948.025

• Transparent reporting form

DOI: https://doi.org/10.7554/eLife.28948.026

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
