## [Decision Letter]

[Editors’ note: this article was originally rejected after discussions between the reviewers, but the authors were invited to resubmit after an appeal against the decision.]

Thank you for submitting your work entitled "Kinesin Motility Driven by Subdomain Dynamics" for consideration by *eLife*. Your article has been favorably evaluated by a Senior Editor and three reviewers, one of whom is a member of our Board of Reviewing Editors. The following individuals involved in review of your submission have agreed to reveal their identity: Charles Vaughn Sindelar (Reviewer #2); Qiang Cui (Reviewer #3).

Our decision has been reached after consultation between the reviewers. Based on these discussions and the individual reviews below, we regret to inform you that your work will not be considered further for publication in *eLife*.

The main concern raised by the reviewers is that the observation of a mobile and unstructured Switch I element in the kinesin motor goes against a number of experimental observations previously reported. Combined with additional concerns surrounding potential artifacts in the simulations, we feel that this study does not provide sufficient evidence to support the claims.

In the extensive online consultation session between the reviewers, a number of additional points were raised that may help you in strengthening the work for submission elsewhere. One of these points was that there may be additional ordered water molecules not present in the simulations that could play a critical role in maintaining the structured nature of Switch I. Another aspect that was discussed is the influence that different force fields may have on the stability of the Switch I element.

*Reviewer #1:*

This manuscript describes the use of molecular dynamics simulations to investigate the dynamics of the switch I loop in kinesin and its conformational transitions throughout the ATPase and translocation cycles. The molecular dynamics interestingly suggest that elements of the kinesin motor domain are highly mobile and that the that plasticity plays a key role in nucleotide hydrolysis and coupling of the liberated energy to mechanical motion. While the technical aspects of the study are impressive with the extent and duration of the simulations, I have significant concerns about the lack of experimental validation of the proposed models. Especially with the observations described of a highly mobile switch I loop which seem to contradict published structural work – and with one of the co-authors being a specialist of exactly the right type of experimental approaches – one would expect strong experimental evidence be presented in the paper.

*Reviewer #2:*

In this manuscript, the authors use molecular dynamics simulations to look at mobile elements in the kinesin motor domain that are involved in nucleotide processing. Based on these simulations, they argue that the switch I loop and other elements close to the nucleotide pocket are much more mobile (particularly in the ATP-bound state of the motor) than was evident in prior structural studies performed with cryo-EM or X-ray crystallography. They further argue that this putative high mobility is relevant to the ATP binding, hydrolysis and product release functions of the motor. They also describe attempts to observe nucleotide binding and unbinding directly via simulation; while neither of these processes completed within the timescale of the reported molecular dynamics runs, some additional claims are made regarding structural elements that may be involved in nucleotide binding/unbinding. At the end of the manuscript a model for the kinesin cycle is presented that closely follows the longstanding 'consensus' model for kinesin motility, although the model does not incorporate (nor is reference made to) recent experimental results indicating that a forward step does not come until after ATP is hydrolyzed.

The main messages in this manuscript are centered around the claim that the switch I nucleotide binding loop is highly mobile, even in the microtubule-bound, ATP-bound state where prior structural work has strongly indicated to the contrary. If true, this would certainly change our understanding of the kinesin mechanism, but it is not clear from the manuscript what the mechanistic purpose would be, for example, of having the nucleotide pocket be so unstable in the presumptive catalytically active state. Such a bold and unexpected claim would seem to require independent validation, ideally from experimental methods. However, the only cited support for the switch I mobility claim in the current manuscript is highly indirect (FRET distance measurements from a recent study by (Muretta, Jun et al. 2015)). The evidence to the contrary, on the other hand, is abundant. For example, there are numerous reported cryo-EM structures of kinesin bound to ATP analogs on microtubules where the switch I loop shows little sign of disorder, going all the way back to Kikkawa's sub-nanometer structure in 2006 (Kikkawa and Hirokawa 2006) and continuing up to recent higher-resolution structures (Atherton, Farabella et al. 2014, Shang, Zhou et al. 2014). Perhaps the most striking cryo-EM structure, although not reported as such, is a near-atomic structure from the Nogales lab of microtubule-attached kinesin with a non-hydrolyzing switch II (E236A) mutation, bound with intact ATP ((Zhang, Alushin et al. 2015); structure can be viewed at EMDB ID 6348); again, there is little to no evidence of switch I mobility in this latter structure.

Moreover, the structure of kinesin's switch I loop has been studied by direct labeling with EPR probes (Naber, Minehardt et al. 2003) and found to be highly immobilized in the microtubule-bound, ATP analog bound states (this work is not cited in the current manuscript, but should be). The X-ray co-crystal structure of a kinesin-tubulin complex bound to ATP analogs (Gigant, Wang et al. 2013) also indicate that the switch I loop is highly ordered, without any indication from B-factors of switch I mobility. The current manuscript tries to argue that switch I in this latter structure has been perturbed by a crystal structure contact, but the associated figure (Figure 5—figure supplement 1) reveals that the contact consists of a single hydrogen bond coming from lysine side chain that extends out from a nearby symmetry mate – I do not find this to be a compelling argument. The switch I loop is already known to be unstable in ATP analog-bound kinesins when tubulin is absent (viz. the previously cited EPR study as well as numerous X-ray studies showing disorder in the presence of ATP analogs; see for example (Nitta, Kikkawa et al. 2004), so it is not surprising to see more extensive crystal contacts stabilizing switch I in the cases where switch I folds to its 'catalytically active' conformation in the absence of tubulin (cf. Figure 5—figure supplement 1). In summary, multiple existing lines of experimental evidence run counter to the authors' claim that switch I is mobile in its catalytically active phase on microtubules.

A major concern here is that the nucleotide pocket in the reported molecular dynamics simulations could be artifactually destabilized by, for example, inaccuracies in the initial conditions or energy function of the simulation. Indeed, the reported simulations seem to indicate that switch I is not only 'mobile' in the ATP-bound, microtubule-bound state, but markedly unstable: it is stated that the switch I loop dissociates from its ATP-coordinating position in multiple simulations, but apparently never re-folds. This feature of the simulation results amplifies the above concerns, because a truly unstable switch I would not show up in experimental density maps (X-ray or EM)- contrary to existing evidence.

The paper makes a number of other potentially interesting claims that are, however, not well substantiated either by the simulations or by prior findings. For example, it is suggested that (1) the initial stages of ATP recruitment might be facilitated by a trio of structural elements of L5, switch I and α 0; or (2) that ADP and Pi products might be 'carried out' of the nucleotide pocket by the switch I loop as a part of nucleotide release. Unfortunately, due to limitations in existing computing power, the timescales of the reported simulations (while impressive) is not sufficient to investigate such claims in detail. The validity of these latter two proposals is also called into question due to the fact that they both involve the switch I loop; the likelihood that the current simulations have problems that may destabilize the structure of switch I (see above) is therefore a significant concern. Moreover, recent findings that the tethered partner head steps only steps forward after hydrolysis, while ADP•Pi is bound in the active site of attached kinesin domain ((Milic, Andreasson et al. 2014, Andreasson, Milic et al. 2015, Mickolajczyk, Deffenbaugh et al. 2015); also reviewed in (Hancock 2016)) seems to indicate that ADP•Pi should persist in kinesin's active site long enough for a successful forward step. In contrast, in the currently reported simulations the active site falls apart (and phosphate dissociates) in less than a microsecond. This seeming discrepancy with the recent experimental evidence is noteworthy, but is never mentioned in the manuscript despite many of the relevant articles having been cited.

I was interested in the observation and discussion of strain in ATP and in kinesin's β sheet. However, these are not strongly emphasized in the manuscript and I am left with questions/concerns, particularly with regards to the β sheet strain. The model for strain is a second-order Gaussian surface, which is characterized by a single curvature parameter. It is not clear to me whether this can accurately capture the 'wrinkle' that appears in kinesin's β sheet when the two halves rotate with respect to each other. The free energy of the strain could therefore be significantly underestimated. For the ATP strain, it is not clear whether the magnitude of the observed strain (1.2kcal/mol) falls in the expected range, or how this would change our view of kinesin's mechanochemistry.

In summary, I do not find the major claims of this paper, regarding the nucleotide pocket and the switch I loop, to be justified by the reported experiments – and indeed I think they likely arise from artifacts related to the simulation methods. The Discussion does not connect the simulation results to the most recent findings regarding kinesin's ADP•Pi state, even where there are seeming contradictions. Some of the other observations in the paper, particularly regarding strain in ATP and/or the β sheet, seem to have more merit. However, on the whole I think this work should be substantially revised and belongs in a more specialized journal.

*Reviewer #3:*

In this molecular dynamics (MD) simulation study of kinesin, the authors conducted a systematic analysis of conformational dynamics of a single kinesin motor domain, either bound to a microtubule (MT) dimer unit or in solution; several nucleotide binding states (ATP, ADP/Pi, ADP and Apo) have also been examined. Compared to previous MD studies, the current work substantially extended the simulation time scale to multiple microseconds (for each state) by taking advantage of resources available on Anton. This is significant since as the authors showed that in most cases, the conformational properties of interest reach a plateau only after about one or a few microsecond(s). The extensive sampling is also essential to a meaningful estimate of the bending energy of the central β-sheet and an evaluation of its relevance to nucleotide processing.

The MD results clearly highlighted that the motor domain is highly dynamical and features rich variations in structural rigidity, hydration and protein-protein as well as protein-nucleotide contacts. These observations provide a rather different description of the kinesin motor domain from static crystal structures and explain the variation of several structural motifs (e.g., SwI) in different crystal structures. The comparison of different nucleotide states also made it possible to reveal specific structural motifs that are likely important to ATP binding, activation of hydrolysis and facilitation of the release of Pi and ADP. The plastic nature of kinesin/MT interface has not been emphasized in previous studies and is likely important to kinesin/MT recognition in vivo. Overall, the results provided new insights into the motor domain properties that are likely important to the mechanochemical coupling in kinesin, and the study has laid the ground work for better understanding communication between two kinesin domains at a molecular level.

[Editors’ note: what now follows is the decision letter after the authors submitted for further consideration.]

Thank you for resubmitting your work entitled "Kinesin Motility Driven by Subdomain Dynamics" for further consideration at *eLife*. Your revised article has been favorably evaluated by Arup Chakraborty (Senior Editor) and three reviewers, one of whom is a member of our Board of Reviewing Editors and another who was not involved with the evaluation of the original manuscript.

The manuscript has been improved considerably and reviewers are satisfied with the way in which you have addressed the points raised during the previous round of review. All reviewers, including a new reviewer, agree that the work described in this paper is of high quality. There remain some technical concerns that will need to be addressed before we can reach a firm decision to accept the paper for publication in *eLife*. These technical points are outlined below:

1) It is stated that kinesin-bound ATP is torsionally strained. There is a comparison with ATP in solution to support this conclusion (For comparison, we performed a 4-ns simulation of an isolated Mg-ATP in water). The authors may wish to have a look at an improved parameterization of the dihedrals of ATP in the CHARMM force field (Komuro et al. JCTC vol 10, 4133-4142, 2014) to verify that this conclusion is robust or force field dependent.

2) The significance of the PMF shown in Figure 2 is not entirely clear. It is stated that "To align PMFs for ATP and APO, we identified bins of the histogram that have nonzero counts in both simulations. We determined the constant free energy shift that needs to be added to the PMF for APO so that the mean-square difference of the two PMFs in the overlapping bins is minimized." The energy barriers on this combined free energy surface are also discussed ("There is also a Ì1.7 kB T energy barrier from ATP towards APO"). In practice, two separate PMFs were extracted from the marginal distributions (-kt*ln(rho)) obtained from two separate simulations that correspond to two very different molecular contexts (one with ATP bound and one without ATP bound). These two PMFs were combined by matching the histograms that have nonzero counts. But that is presuming that the two surfaces exist with respect to the same part of configuration space. Since one of them has a ligand bound and the other does not, why should this be so? Unless we misunderstand, the configurational sampling of the two state do not overlap, so how can the two PMFs be combined? In principle, the relative shift between two free energy surfaces, one ligand-bound and one APO, should depend on the concentration of the ligand. If the ligand concentration is zero, then the PMF of the ligand-bound state shifts to infinity, and likewise, if the ligand concentration is infinite then the PMF of the APO state shifts to infinity. This point needs to be addressed and clarified.

3) In Figure 3, the binding energies are of the order of -100 to -250 kcal/mol. These values appear to be very large, and are probably not standard binding free energies. The Materials and methods section states that "Calculation of the binding energy was done based on a previously developed method (Zoete et al., 2005)." This method is MM/GB-SA (molecular mechanics-generalized Born surface area) which is an end-point approximation to the binding free energy. There is scepticism that the binding energy estimated by the Generalized Born method is sufficiently accurate to draw meaningful conclusions for the system being considered. How sensitive are the conclusions to this analysis?

---

## [Author Response]

Editors’ note: the author responses to the first round of peer review follow.]

Reviewer #1:This manuscript describes the use of molecular dynamics simulations to investigate the dynamics of the switch I loop in kinesin and its conformational transitions throughout the ATPase and translocation cycles. The molecular dynamics interestingly suggest that elements of the kinesin motor domain are highly mobile and that the that plasticity plays a key role in nucleotide hydrolysis and coupling of the liberated energy to mechanical motion. While the technical aspects of the study are impressive with the extent and duration of the simulations, I have significant concerns about the lack of experimental validation of the proposed models. Especially with the observations described of a highly mobile switch I loop which seem to contradict published structural work – and with one of the co-authors being a specialist of exactly the right type of experimental approaches – one would expect strong experimental evidence be presented in the paper.

We thank reviewer 1 for recognizing the technical strength and novelty of our work. The consistency of sw-I’s (partial) mobility with existing data is detailed in our response to comments by reviewer 2 below, who had more specific points. Our simulation agrees with the available experimental data, as explained above and throughout the manuscript. Given that the proposed mechanism is novel, we believe it is important to publish it, as a prediction that we hope will stimulate further experiments.

Change made:

Subsection “Concluding Discussion”, last paragraph: Need for future experiments is explained, in light of the conceptual advances made in this work.

Reviewer #2:In this manuscript, the authors use molecular dynamics simulations to look at mobile elements in the kinesin motor domain that are involved in nucleotide processing. Based on these simulations, they argue that the switch I loop and other elements close to the nucleotide pocket are much more mobile (particularly in the ATP-bound state of the motor) than was evident in prior structural studies performed with cryo-EM or X-ray crystallography. They further argue that this putative high mobility is relevant to the ATP binding, hydrolysis and product release functions of the motor. They also describe attempts to observe nucleotide binding and unbinding directly via simulation; while neither of these processes completed within the timescale of the reported molecular dynamics runs, some additional claims are made regarding structural elements that may be involved in nucleotide binding/unbinding. At the end of the manuscript a model for the kinesin cycle is presented that closely follows the longstanding 'consensus' model for kinesin motility, although the model does not incorporate (nor is reference made to) recent experimental results indicating that a forward step does not come until after ATP is hydrolyzed.

We reply to the comments about the mobility of subdomains in other replies below. In referring to a kinesin dimer, in the caption of Figure 8, we do make reference to the paper that reviewer 2 mentioned regarding “recent experimental results indicating that a forward step does not come until after ATP is hydrolyzed” (Milic et al., 2014). But we mistakenly stated that P*_i_*release (instead of ATP hydrolysis) completes a step. This is corrected in the revision. However, this does not affect our main findings, which concern the nucleotide processing mechanisms of a single motor head. Figure 8 demonstrates how the dynamic mechanism may be at work in the dimer motility. In reviewer 3’s second point, reviewer 3 recognizes this: “the results provided new insights into the motor domain properties that are likely important to the mechanochemical coupling in kinesin, and the study has laid the ground work for better understanding communication between two kinesin domains at a molecular level.” We clarified these points in the revised manuscript.

While the motility of a kinesin dimer is not the main focus of the present work, the model of dimer motility based on subdomain dynamics provides a possible explanation for how P*_i_*release triggers detachment of the rear head, which directly aligns with the finding by Milic et al.(2014): With P*_i_*in place, conformational transition of *α*4 that leads to detachment of the motor head (Figure 8) is suppressed, so that P*_i_*release is required for unbinding of kinesin from the MT.

Changes made:

Introduction, second paragraph, Figure 8, and its caption are corrected, to be consistent with the model proposed in Milic et al.(2014). Video 5 is also updated.

Subsection “Concluding Discussion”, second paragraph: Scope of the present work is explained (please also see the second paragraph of the Introduction).

Subsection “Concluding Discussion”, fifth paragraph: Proposal in Milic et al. (2014) regarding the motor head detachment upon P*_i_*release, is explained in terms of our dynamical picture.

The main messages in this manuscript are centered around the claim that the switch I nucleotide binding loop is highly mobile, even in the microtubule-bound, ATP-bound state where prior structural work has strongly indicated to the contrary. If true, this would certainly change our understanding of the kinesin mechanism, but it is not clear from the manuscript what the mechanistic purpose would be, for example, of having the nucleotide pocket be so unstable in the presumptive catalytically active state. Such a bold and unexpected claim would seem to require independent validation, ideally from experimental methods. However, the only cited support for the switch I mobility claim in the current manuscript is highly indirect (FRET distance measurements from a recent study by (Muretta, Jun et al. 2015)). The evidence to the contrary, on the other hand, is abundant. For example, there are numerous reported cryo-EM structures of kinesin bound to ATP analogs on microtubules where the switch I loop shows little sign of disorder, going all the way back to Kikkawa's sub-nanometer structure in 2006 (Kikkawa and Hirokawa 2006) and continuing up to recent higher-resolution structures (Atherton, Farabella et al. 2014, Shang, Zhou et al. 2014). Perhaps the most striking cryo-EM structure, although not reported as such, is a near-atomic structure from the Nogales lab of microtubule-attached kinesin with a non-hydrolyzing switch II (E236A) mutation, bound with intact ATP ((Zhang, Alushin et al. 2015); structure can be viewed at EMDB ID 6348); again, there is little to no evidence of switch I mobility in this latter structure.

As mentioned at the beginning of this reply, unfolding of the sw-I hairpin in the ATP state is only partial: Its N-terminal side breaks backbone hydrogen bonds with the C-terminal side of sw-I (R190–T195 in Figure 5). However, the C-terminal side containing the SSR motif maintains catalytically important contacts with ATP and the sw-II domain throughout the simulation (subsection “Catalytic water molecules are dynamically coordinated”, first paragraph). Thus, reviewer 2’s remark, “the nucleotide pocket be so unstable in the presumptive catalytically active state,” is inapplicable.

To compare the observed mobility of sw-I with cryo-EM structures that reviewer 2 mentioned, we docked the structures before and after unfolding of the sw-I hairpin. We only considered recent high resolution structures of Kinesin-1, and did not consider Kikkawa’s structure for Kinesin-3 (KIF1A). For an unbiased comparison, instead of performing flexible fitting, we rigidly docked atomic structures to the cryo-EM map using the stably folded central *β*-sheet and *α*4 as reference. We added this as the new Figure 6. Even for the structure with intact hairpin (yellow in Figure 6), the N-terminal side of sw-I does not match well with the EM density compared to the C-terminal side which firmly contacts ATP (Figure 6, arrows). Given the anchoring of the C-terminal side, the motion of the N-terminal side of sw-I is limited. As Figure 2 vs. E shows, the unfolded state of sw-I is closer to the hairpin conformation compared to the case of an isolated kinesin in the ATP state. At cryogenic temperatures, the N-terminal side is thus likely to settle to a hairpin-like state with the C-terminal side as a template, instead of landing in different configurations that lead to low electron density.

In cryo-EM experiments, in addition to the low temperature, MTs are densely covered with kinesins, so that kinesins would be conformationally more restricted than when individual motors walk on the MT at room or body temperature. Furthermore, kinesins can hydrolyze ATP when they are bound to unpolymerized tubulin dimers, with catalytic rates ranging from lower than to comparable to those bound to MT filaments (Alonso, et al., 2007). Even an isolated kinesin can hydrolyze ATP, though with a lower rate (Introduction, third paragraph). We also demonstrate that even when the N-terminal side of sw-I is detached from the C-terminal side, catalytic water molecules and contacts between sw-I and ATP/sw-II are maintained, which are sufficient for hydrolyzing an ATP. Thus, the “mechanistic purpose” of the flexibility of sw-I would be to fine tune kinetic rates. Since the residues of sw-I contacting ATP and sw-II are highly conserved, they by themselves cannot determine family-specific catalytic rates. By modifying flexibility of the distal domain, the condition under which the hydrolysis reaction occurs can be controlled. We also note that, the hydrolysis reaction occurs over a picosecond time scale, as ab initiocalculations show (McGrath et al., 2013). Even though a fully folded hairpin state of sw-I may support hydrolysis more efficiently, it does not need to stay stably folded throughout the ATP state. Cao et al.(2014) shows that the R190A/D231A mutant has the hydrolysis rate reduced to 22% of the wild-type value. As R190 is at the N-terminus of sw-I, breaking this contact will destabilize the hairpin, but it does not abolish ATP hydrolysis (see reply to reviewer 2’s next pointbelow). This further supports that the propensity for forming hairpin can affect the hydrolysis rate, but it is not required. Thus, the x-ray and cryo-EM data, which are static in nature, provide the foundation for developing the dynamic picture in our study, rather than contradicting it.

As to the FRET paper by Muretta et al. (2015), reviewer 2 states that the measurement is “highly indirect.” However, a main focus of the paper is the mobility of sw-I, as its title suggests: “The structural kinetics of switch-1 and the neck linker explain the functions of kinesin-1 and Eg5.” As explained in our manuscript (subsection “Concluding Discussion”, second paragraph), their finding that sw-I stays ‘open’ in significant mole fraction, even in the ATP state, is consistent with our picture. But we also agree with reviewer 2 that experiments probing the flexibility of sw-I are scant, perhaps because the dynamic nature of sw-I has not been recognized. Since our work is the first systematic study, we hope that our work will lead to further experimental studies probing the dynamic aspect.

Changes made:

Added Figure 6 that compares cryo-EM structures and structures before and after unfolding of the sw-I hairpin, together with discussions in the last paragraph of the subsection “Sw-I hairpin unfolds”. For Figure 6, Nogales’ paper (Zhang et al., 2015) is cited – we thank the reviewer for pointing out this impressive work.

Subsections “Nucleotide pocket experiences large changes in intra-kinesin contacts”; “Sw-I hairpin unfolds”, second paragraph; “Catalytic water molecules are dynamically coordinated”, first paragraph: Explanations about the partial nature of the sw-I unfolding are given.

Subsection “Concluding Discussion”, second and third paragraphs: Expanded discussion about the flexibility of sw-I, including the partial nature of unfolding, mechanistic purpose of fine tuning, works by Alonso, et al. (2007), and the behavior of the R190A/D231A mutant.

Moreover, the structure of kinesin's switch I loop has been studied by direct labeling with EPR probes (Naber, Minehardt et al. 2003) and found to be highly immobilized in the microtubule-bound, ATP analog bound states (this work is not cited in the current manuscript, but should be). The X-ray co-crystal structure of a kinesin-tubulin complex bound to ATP analogs (Gigant, Wang et al. 2013) also indicate that the switch I loop is highly ordered, without any indication from B-factors of switch I mobility. The current manuscript tries to argue that switch I in this latter structure has been perturbed by a crystal structure contact, but the associated figure (Figure 5—figure supplement 1) reveals that the contact consists of a single hydrogen bond coming from lysine side chain that extends out from a nearby symmetry mate – I do not find this to be a compelling argument. The switch I loop is already known to be unstable in ATP analog-bound kinesins when tubulin is absent (viz. the previously cited EPR study as well as numerous X-ray studies showing disorder in the presence of ATP analogs; see for example (Nitta, Kikkawa et al. 2004), so it is not surprising to see more extensive crystal contacts stabilizing switch I in the cases where switch I folds to its 'catalytically active' conformation in the absence of tubulin (cf. Figure 5—figure supplement 1). In summary, multiple existing lines of experimental evidence run counter to the authors' claim that switch I is mobile in its catalytically active phase on microtubules.

The EPR measurement of Naber et al.(2003) shows that sw-I changes from an open to a closed state when kinesin binds to the MT in the presence of ATP, and it does not address the nucleotide-dependent conformational transition of sw-I in a MT-bound kinesin. As shown in Figure 2, as well as in the displacement/RMSD data, sw-I is deformed more from the initial hairpin conformation in *Kin-only* than in *ATP*. Instead of contradicting the EPR measurement, our simulation is thus consistent with it.

For the x-ray structure with ATP-analog in Nitta et al. (2004) that reviewer 2 pointed out (PDB 1VFV), we constructed the corresponding crystal and added it as Figure 5—figure supplement 1. Sw-I in this case is only partially visible. As expected, it does not have any crystal contact. Notably, the C-terminal side of sw-I including the SSR motif forms contact with the nucleotide, whereas the partially visible N-terminal side is detached from the C-terminal side (shown on the right side of Figure 5—figure supplement 1). This is entirely consistent with our observation in *Kin-only* simulation. We have verified similar behaviors of partially visible sw-I in other x-ray structures of kinesin in ATP-analog states listed in Supplementary file 1.

Regarding the crystal contact in PDB 4HNA (Figure 5—figure supplement 1), the lysine side chain of the neighboring complex makes contact with the backbone oxygen of R190, which is located at the N-terminal end of sw-I. Since the sw-I hairpin opens in a zipper-like manner starting from R190 (Figure 5), constraining it there should be sufficient to stabilize the entire hairpin. For other crystal contacts, although reviewer 2 said “The switch I loop is already known to be unstable in ATP analog-bound kinesins when tubulin is absent,” we feel it is important to explicitly show these contacts, especially since our simulation demonstrates the flexibility of sw-I.

Regarding reviewer 2’s summary, we again emphasize that, unfolding is only partial, and our results elucidate dynamic aspects of kinesin as an ATPase without contradicting previous experiments.

Changes made:

Abstract: Changed “mobile switch domains” to “switch domains,” to avoid confusion from the beginning.

Subsection “Functionally important subdomains are mobile”, second paragraph: Added the sequence range of sw-I, for clarity.

Figure 2, caption: Changed “Sw-I is unfolded” to “Sw-I lost its pseudo *β*-hairpin conformation,” to be more specific.

Results: Changed section title from “Sw-I hairpin is inherently unstable,” to “Sw-I hairpin unfolds,” to avoid giving the impression that the entire sw-I including its C-terminal side is unstable.

Subsection “Sw-I hairpin unfolds”, second paragraph: Conformation of sw-I in Nitta’s structure is explained. Added Figure 5—figure supplement 1, and cited Nitta et al. (2004).

Figure 5—figure supplement 1 caption: Explained stabilization of R190 at the N-terminus of sw-I via crystal contact.

Subsection “Concluding Discussion”, second paragraph: Included discussion about Naber’s EPR experiment, and also cited Naber et al.(2003).

A major concern here is that the nucleotide pocket in the reported molecular dynamics simulations could be artifactually destabilized by, for example, inaccuracies in the initial conditions or energy function of the simulation. Indeed, the reported simulations seem to indicate that switch I is not only 'mobile' in the ATP-bound, microtubule-bound state, but markedly unstable: it is stated that the switch I loop dissociates from its ATP-coordinating position in multiple simulations, but apparently never re-folds. This feature of the simulation results amplifies the above concerns, because a truly unstable switch I would not show up in experimental density maps (X-ray or EM)- contrary to existing evidence.

Through many tests that we performed, we are confident that unfolding of the sw-I hairpin is not a simulation artifact. The unfolding also occurs in currently ongoing simulations of a kinesin dimer on the MT protofilament (not included in our manuscript). Our careful simulation preparation and execution are also recognized by reviewer 3: “The work has been rigorously conducted and analyzed.” Neither is it likely that the unfolding of sw-I is due to the inaccuracy of the force field, since many other aspects of our simulation are consistent with previous experimental data. The force field for ATP cannot be an issue either, since contacts between ATP with the P-loop and the C-terminal part of sw-I were maintained throughout simulations, and ATP does not have any direct contact with the N-terminal part of sw-I. Moreover, the CHARMM param36 force field used in this study is presently the most accurate available, and it has been successfully used in many other multi-microsecond simulations on the Anton supercomputer.

While reviewer 2 mentions that sw-I is “markedly unstable,” stability has a relative meaning. Since the sw-I hairpin unfolding occurred only after at least several hundred nanoseconds from the start of our simulation, it may not be observed in shorter simulations. For example, Chakraborty and Zheng (2015) performed a 400-ns all-atom simulation of a kinesin-MT complex in the ATP state, and did not report unfolding of sw-I. Notably, the 3–4-˚A RMSF of sw-I in their simulation (Figure 2 of Chakraborty and Zheng, 2015) is *larger* than that during the first 400 ns of our *ATP* simulation (less than 2.1 ˚A; Figure 2 of our manuscript), which further supports that our simulation system was well-prepared. Although not discussed, the RMSF analysis in Chakraborty and Zheng (2015) also indicates that sw-I is among the most fluctuating subdomains of the motor head, which is consistent with our result.

As explained in Figure 5, refolding of sw-I is hampered by the outward rotation of *α*3, which keeps the two termini of sw-I apart. Its refolding is thus not simply a local event, but it will involve the conformational motion of the entire motor head that allows inward rotation of *α*3. Such global-level changes would occur on a time scale longer than that of our simulation, which may be why refolding of the sw-I hairpin was not observed. However, the rate of folding/unfolding of the sw-I hairpin, although it might affect the ATP hydrolysis rate, does not affect our main finding, since we have clearly shown that sw-I can support ATP hydrolysis as long as its C-terminal side maintains proper contacts with ATP and sw-II (Figure 5).

Regarding reviewer 2’s remark, “it is stated that the switch I loop dissociates from its ATP-coordinating position,” we *did not*make that statement – even after the hairpin unfolds, catalytically important contacts involving the C-terminal part of sw-I and coordination of catalytic water molecules, are maintained. As explained in response to reviewer 2’s second and third pointsabove, we believe that the revised manuscript clarifies the partial nature of unfolding and its consistency with previous experimental data.

Changes made:

Subsection “Functionally important subdomains are mobile”, second paragraph: Consistency of our result with the mobility of sw-I in Chakraborty and Zheng (2015) is explained.

Subsection “Functionally important subdomains are mobile”, second paragraph: Explains the slow time scale associated with the refolding of the sw-I hairpin.

The paper makes a number of other potentially interesting claims that are, however, not well substantiated either by the simulations or by prior findings. For example, it is suggested that (1) the initial stages of ATP recruitment might be facilitated by a trio of structural elements of L5, switch I and α 0; or (2) that ADP and Pi products might be 'carried out' of the nucleotide pocket by the switch I loop as a part of nucleotide release. Unfortunately, due to limitations in existing computing power, the timescales of the reported simulations (while impressive) is not sufficient to investigate such claims in detail. The validity of these latter two proposals is also called into question due to the fact that they both involve the switch I loop; the likelihood that the current simulations have problems that may destabilize the structure of switch I (see above) is therefore a significant concern. Moreover, recent findings that the tethered partner head steps only steps forward after hydrolysis, while ADP•Pi is bound in the active site of attached kinesin domain ((Milic, Andreasson et al. 2014, Andreasson, Milic et al. 2015, Mickolajczyk, Deffenbaugh et al. 2015); also reviewed in (Hancock 2016)) seems to indicate that ADP•Pi should persist in kinesin's active site long enough for a successful forward step. In contrast, in the currently reported simulations the active site falls apart (and phosphate dissociates) in less than a microsecond. This seeming discrepancy with the recent experimental evidence is noteworthy, but is never mentioned in the manuscript despite many of the relevant articles having been cited.

Observing a complete ATP binding event in an unbiased simulation is beyond the present-day computing power. However, physical factors associated with ATP binding can be found by observing how kinesin interacts with a free ATP without needing to simulate the complete binding process. For this, our simulation of ATP binding provides the following information: 1) The trio domains (*α*0, L5 and sw-I) surrounding the P-loop are mobile (Figure 2; their mobility was also observed by their large RMSF in Figure 2 of Chakraborty and Zheng, 2015); 2) Since the trio domains are located further on the outer side of the motor head than the P-loop, when an ATP approaches, it will first interact with the trio (Figure 5—figure supplement 2); 3) Since ATP is amphiphilic, it is more likely to approach kinesin with the adenosine ring pointing inward while the more hydrophilic triphosphate moiety points outward (Figure 5). It is thus expected that the trio domains will play important roles for binding of an ATP in a proper orientation. Also, reviewer 2’s concern about the mobility of sw-I was for the hairpin conformation in the ATP state, and it does not apply to the APO-state kinesin, whose sw-I is known to be mobile from previous experiments (e.g., it is invisible in PDB 4LNU, used for the *APO* simulation) as well as our present simulation. Longer simulation trajectory would of course be useful for providing more details (e.g., how ATP orients, or how its phosphate moiety binds to the P-loop), but the present simulation was sufficient to establish the basic physical picture. The role of L5 for ATP binding has previously been considered (Atherton et al.2014). However, to our knowledge, the functional roles of the mobile sw-I and *α*0, and the concerted role of the trio domains, are new in the present study. We also would like to add that, though our simulation times are too short for observing the full ATP binding event, their lengths are unprecedented.

Reviewer 2 questions the proposed P*_i_*release mechanism because of the possibility that the sw-I hairpin may have been made unstable in our simulations. As explained above, not only is this possibility unlikely, but P*_i_*release occurred with an intact hairpin (Figure 5 and Figure 5—figure supplement 2). Unfolding of the sw-I hairpin is thus irrelevant. Since the hairpin is by itself mobile (high RMSD during the first 400 ns; Figure 2—figure supplement 1), and since it maintains contacts with P*_i_*, as sw-I moves, it is expected pull P*_i_*.

Another concern by reviewer 2 is that the sub-microsecond release time of P*_i_*in our simulation may be too short compared to the longer lifetime of the ADP·P*_i_*state indicated in recent experiments. But the divalent form of phosphate HPO42-;Pi2- did not release in our simulations, and it may persist in reality (subsection “Release of hydrolysis products is mediated by the mobile sw-I”, second paragraph). Even for the monovalent P*_i_*, in our recent simulations (not included in our original submission), we found that the propensity for its release depends on its orientation. Author response image 1 reproduces the inset of Figure 5, right before the release of P*_i_*. In this case, a hydrogen atom of P*_i_*makes contact with an O*_β_*atom of ADP. Due to the low charge of this hydrogen (0.33*e*), the contact is readily broken, leading to the release of P*_i_*. However, we recently found that P*_i_*can tumble, and its oxygen atom can form a contact with Mg^2+^. In this case, P*_i_*did not release till the end of a 2-*µ*s simulation (Author response image 1). This is due to the stronger electrostatic interaction between the phosphate oxygen (−0.63*e*) and Mg^2+^. To estimate the release time, an ensemble of different release pathways must be considered, which is not the focus of the present study. Regardless of the rate of release, it is physically likely that P*_i_*release is assisted by the mobile sw-I.

**Author response image 1. respfig1:** Influence of P*_i_*orientation on its release. (**A**) Re-rendering of Figure 5, 622.80 ns. P*_i_*does not have a direct contact with Mg^2+^. (**B**) The last frame of a new 2-*µ*s simulation. An oxygen atom of P*_i_*maintains contact with Mg^2+^(marked by a bent line).

For the simulation of ADP release, we used sw-I in an *α*-helical conformation. Given its mobility, it is more plausible that it contributes to the release of ADP, for otherwise ADP will have to break its contacts with the P-loop spontaneously, which will be energetically less favorable. Overall, the dynamic roles of sw-I and other subdomains surrounding the ATP pocket make physical sense, and converse scenarios where sw-I remains static through these processes, are less probable.

Changes made:

Abstract: The total simulation time is updated, to account for the new 2-*µ*s simulation mentioned in the second paragraph of the subsection “Release of hydrolysis products is mediated by the mobile sw-I”.

Subsection “Binding of ATP is mediated by the *α*_0_/L5/sw-I trio”, last paragraph: Physical plausibility of the dynamic role of the trio domains for ATP binding is explained.

Subsection “Release of hydrolysis products is mediated by the mobile sw-I”, second paragraph: The issue of the lifetime of the *ADP+P_i_*state is mentioned with regard to the recent findings in the references that reviewer 2 mentioned. Citation for Mickolajczyk et al.(2015) is added.

Subsection “Release of hydrolysis products is mediated by the mobile sw-I”, second paragraph: The orientation-dependent lifetime of a bound P*_i_*is added. However, Author response image 1 above is not included, since the situation is similar to Figure 5—figure supplement 2.

I was interested in the observation and discussion of strain in ATP and in kinesin's β sheet. However, these are not strongly emphasized in the manuscript and I am left with questions/concerns, particularly with regards to the β sheet strain. The model for strain is a second-order Gaussian surface, which is characterized by a single curvature parameter. It is not clear to me whether this can accurately capture the 'wrinkle' that appears in kinesin's β sheet when the two halves rotate with respect to each other. The free energy of the strain could therefore be significantly underestimated. For the ATP strain, it is not clear whether the magnitude of the observed strain (1.2kcal/mol) falls in the expected range, or how this would change our view of kinesin's mechanochemistry.

We first address reviewer 2’s question, whether the 1.2 kcal/mol torsional energy “falls in the expected range.” Its smallness compared to the ∼10 kcal/mol free energy of ATP hydrolysis indicates that torsional strain readily develops as ATP binds to kinesin. Yet, this primes the scissile bond for attack by the catalytic water (subsection “Kinesin-bound ATP is torsionally strained”, last paragraph). Thus, energetics of ATP hydrolysis is determined as a combined effect among ATP geometry, catalytic water coordination, configurations of the immediately surrounding residues as well as allosteric behaviors of remote parts of the protein. Thanks to reviewer 3’s suggestions, we have now expanded the discussion on both bond elongation and torsional strain in ATP.

Regarding the twist of the central *β*-sheet, instead of a Gaussian surface that reviewer 2 referred to (*e^ax^*^2+*bxy*+*cy*2^), we used a quadratic surface (Equation 1). Perhaps confusion arose because Figure 2 was taken to depict the surface geometry. But as its axis labels and caption indicate, it plots distribution (free energy) of two curvature values. From this, it is also evident that, instead of “a single curvature parameter,” two curvature values are used, the mean and Gaussian curvatures. They respectively describe bending and twisting of curved surfaces in differential geometry. The ‘*wrinkle*’ that reviewer 2 describes is about twisting (“two halves rotate with respect to each other”), hence it is captured by Gaussian curvature. Indeed, Figure 2 and Figure 2—figure supplement 1 show that Gaussian curvature varies more between pre- and post-stroke states compared to mean curvature. To our knowledge, our work is the first to quantitatively measure the curvature energy of the central *β*-sheet. It turns out to be small between the pre- and post-stroke states. Although further studies are warranted, we believe that our analyses of the strain in ATP and the curvature of the central *β*-sheet clearly advance our understanding of kinesin’s mechanochemistry.

Changes made:

Subsection “Kinesin’s central *β*-sheet does not store enough energy to drive nucleotide processing”, second paragraph: Meaning of mean and Gaussian curvatures as descriptors of the bending and twisting of the *β*-sheet is explained.

We did not add any additional explanation about the curvature of the central *β*-sheet, since we believe existing explanations (subsection “Kinesin’s central *β*-sheet does not store enough energy to drive nucleotide processing”, second and last paragraphs; subsection “Concluding Discussion”, eighth paragraph) are sufficient.

Subsection “Kinesin-bound ATP is torsionally strained”, last paragraph: Implication of the smallness of the torsional energy in kinesin-bound ATP is explained.

Subsection “Kinesin-bound ATP is torsionally strained”, last paragraph: Comparison with the torsional strain of ATP in myosin, and additional discussion about its effect on ATP hydrolysis are provided.

In summary, I do not find the major claims of this paper, regarding the nucleotide pocket and the switch I loop, to be justified by the reported experiments – and indeed I think they likely arise from artifacts related to the simulation methods. The Discussion does not connect the simulation results to the most recent findings regarding kinesin's ADP•Pi state, even where there are seeming contradictions. Some of the other observations in the paper, particularly regarding strain in ATP and/or the β sheet, seem to have more merit. However, on the whole I think this work should be substantially revised and belongs in a more specialized journal.

Through our replies above, we believe that we have sufficiently addressed reviewer 2’s concerns about the mobility of sw-I, as well as the reviewer’s impression that experimental data are inconsistent with our results. We appreciate that reviewer 2 recognizes merits in our analyses of the strain in ATP and the central *β*-sheet. Given that the proposed mechanisms are novel, and that they have broad relevance to other kinesin families and nucleotide triphosphatases, we believe *eLife* is an ideal place to publish this work.

Reviewer #3:In this molecular dynamics (MD) simulation study of kinesin, the authors conducted a systematic analysis of conformational dynamics of a single kinesin motor domain, either bound to a microtubule (MT) dimer unit or in solution; several nucleotide binding states (ATP, ADP/Pi, ADP and Apo) have also been examined. Compared to previous MD studies, the current work substantially extended the simulation time scale to multiple microseconds (for each state) by taking advantage of resources available on Anton. This is significant since as the authors showed that in most cases, the conformational properties of interest reach a plateau only after about one or a few microsecond(s). The extensive sampling is also essential to a meaningful estimate of the bending energy of the central β-sheet and an evaluation of its relevance to nucleotide processing.

Change made:

Subsection “Concluding Discussion”, last paragraph: Significance of performing multi-microsecond simulations for obtaining convergent behaviors is explained.

[Editors’ note: the author responses to the re-review follow.]

The manuscript has been improved considerably and reviewers are satisfied with the way in which you have addressed the points raised during the previous round of review. All reviewers, including a new reviewer, agree that the work described in this paper is of high quality. There remain some technical concerns that will need to be addressed before we can reach a firm decision to accept the paper for publication in eLife. These technical points are outlined below:1) It is stated that kinesin-bound ATP is torsionally strained. There is a comparison with ATP in solution to support this conclusion (For comparison, we performed a 4-ns simulation of an isolated Mg-ATP in water). The authors may wish to have a look at an improved parameterization of the dihedrals of ATP in the CHARMM force field (Komuro et al. JCTC vol 10, 4133-4142, 2014) to verify that this conclusion is robust or force field dependent.

Upon the reviewer’s suggestion, we used the modified ATP force field from Komuro’s paper and re-calculated the dihedral energy of ATP. In the table below, energies are in kcal/mol units (avg ± std):

Force Field*ATP-only* (in solution)*ATP* (bound to kinesin)Original (param36)28.48 ± 2.3829.63 ± 1.85Modified (Komuro)28.11 ± 2.3727.59 ± 1.86

Between *ATP-only* (isolated ATP) and *ATP* (kinesin-bound), the energy decreases slightly with the modified parameters whereas it increases with the original parameters. As explained in Komuro’s paper, the modified parameters render ATP more flexible, so that its conformation is more stable in certain structures with a bound ATP. However, they also state that the original parameters “have worked reasonably well in simulations performed on many ATP-bound proteins.” They also note: “There seems to be intrinsic difficulty in reproducing all the properties with existing force-field parameters, and different applications may need different adjustments.” Regardless of which parameters are more appropriate for our system, the conclusion that the dihedral energy change does not make a significant contribution, is still valid.

Change made:added:

“[…]torsional angles of the phosphate moiety should readily change when ATP binds to kinesin (calculation using a modified force field that worked well for certain ATP-bound protein structures (Komuro et al., 2014), also yielded only marginal changes in the dihedral energy).”

2) The significance of the PMF shown in Figure 2 is not entirely clear. It is stated that "To align PMFs for ATP and APO, we identified bins of the histogram that have nonzero counts in both simulations. We determined the constant free energy shift that needs to be added to the PMF for APO so that the mean-square difference of the two PMFs in the overlapping bins is minimized." The energy barriers on this combined free energy surface are also discussed ("There is also a Ì1.7 kB T energy barrier from ATP towards APO"). In practice, two separate PMFs were extracted from the marginal distributions (-kt*ln(rho)) obtained from two separate simulations that correspond to two very different molecular contexts (one with ATP bound and one without ATP bound). These two PMFs were combined by matching the histograms that have nonzero counts. But that is presuming that the two surfaces exist with respect to the same part of configuration space. Since one of them has a ligand bound and the other does not, why should this be so? Unless we misunderstand, the configurational sampling of the two state do not overlap, so how can the two PMFs be combined? In principle, the relative shift between two free energy surfaces, one ligand-bound and one APO, should depend on the concentration of the ligand. If the ligand concentration is zero, then the PMF of the ligand-bound state shifts to infinity, and likewise, if the ligand concentration is infinite then the PMF of the APO state shifts to infinity. This point needs to be addressed and clarified.

We thank the reviewer for the insightful comment. The reviewer correctly points out that Figure 2 is not the potential of mean force (PMF) from the simulation of an alchemical system where the Hamiltonian contains both ATP and APO states. The two curvature values as reaction coordinates, do not describe other conditions in individual simulations either (such as the presence or absence of ATP). However, as Figure 2—figure supplement 1 shows, PMFs for individual simulations are similar for a given conformational state of the motor head (pre- or poststroke), and they do not depend directly on the bound nucleotide (ATP or ADP+P*_i_*) or whether or not kinesin is bound to the MT. The large overlap in the curvature distributions between the pre- and post-stroke states (Figure 2—figure supplement 1) indicates that the central *β*-sheet cannot store sufficient energy for driving the motility cycle. In this regard, the ‘combined’ PMF in Figure 2 is merely a device that provides a semi-quantitative picture of free energy differences. Even if an alchemical simulation were performed (which would be difficult in practice), the free energies are unlikely to be very different, and our conclusion that the curvature energy does not change significantly between the pre- and post-stroke states, should hold.

Changes made:

Subsection “Kinesin’s central *β*-sheet does not store enough energy to drive nucleotide processing”, second paragraph: The abbreviation ‘PMF’ is introduced.

Subsection “Kinesin’s central *β*-sheet does not store enough energy to drive nucleotide processing”, last paragraph: The nature of the PMF in Figure 2 and its relation to Figure 2—figure supplement 1, are explained.

3) In Figure 3, the binding energies are of the order of -100 to -250 kcal/mol. These values appear to be very large, and are probably not standard binding free energies. The Materials and methods section states that "Calculation of the binding energy was done based on a previously developed method (Zoete et al., 2005)." This method is MM/GB-SA (molecular mechanics-generalized Born surface area) which is an end-point approximation to the binding free energy. There is scepticism that the binding energy estimated by the Generalized Born method is sufficiently accurate to draw meaningful conclusions for the system being considered. How sensitive are the conclusions to this analysis?

As explained in Zoete’s paper, rather than the issue of the accuracy of the Generalized Born method, the binding energy between proteins without considering entropy tends to be larger than the experimental binding free energy. The binding free energy consists of respectively large enthalpy and entropy terms that subtract to yield experimental values. In Zoete’s work, the binding energy for an insulin dimer was −39 kcal/mol, where an insulin monomer is 51 residues in size. As a kinesin motor head is more than 6 times larger, the calculated binding energy at about −240 kcal/mol, as in Figure 3 in our previous submission, is reasonable. However, although the differences in the binding energies are consistent with experimental observations and with our other analyses regarding the kinesin-MT interface (subsection “Kinesin-MT interface is dynamic and hydrated”, last paragraph and subsection “Kinesin-MT interface is dynamic and hydrated”, first paragraph), we agree that only considering the binding energy (enthalpy) is not sufficient for a quantitative comparison with experiment. We thus explained limitations of this calculation and moved the figure to Figure 3—figure supplement 1. Given the large size of the system, calculation of the full binding free energy would be the subject of another project.

Changes made:

Moved Figure 3 to Figure 3—figure supplement 1, and updated the caption.

Modified the sentence:

“However, our binding energies do not include water-mediated interactions and entropic contributions, which are expected to be comparable to the binding energy in magnitude (Zoete et al., 2005), so that the net binding free energy is much smaller than those in Figure 3—figure supplement 1. Thus, the calculated binding energies, although they reflect the interaction between kinesin and the MT in different nucleotide states, do not correspond quantitatively to the experimental binding affinities.”